# The transcriptional corepressor CtBP2 serves as a metabolite sensor orchestrating hepatic glucose and lipid homeostasis

Motohiro Sekiya [1✉], Kenta Kainoh[1], Takehito Sugasawa[1], Ryunosuke Yoshino[2], Takatsugu Hirokawa[2], Hiroaki Tokiwa[3], Shogo Nakano [4], Satoru Nagatoishi [5], Kouhei Tsumoto [5,6], Yoshinori Takeuchi[1], Takafumi Miyamoto[1], Takashi Matsuzaka[1,2] & Hitoshi Shimano [1]

Biological systems to sense and respond to metabolic perturbations are critical for the maintenance of cellular homeostasis. Here we describe a hepatic system in this context orchestrated by the transcriptional corepressor C-terminal binding protein 2 (CtBP2) that harbors metabolite-sensing capabilities. The repressor activity of CtBP2 is reciprocally regulated by NADH and acyl-CoAs. CtBP2 represses Forkhead box O1 (FoxO1)-mediated hepatic gluconeogenesis directly as well as Sterol Regulatory Element-Binding Protein 1 (SREBP1)-mediated lipogenesis indirectly. The activity of CtBP2 is markedly defective in obese liver reflecting the metabolic perturbations. Thus, liver-specific CtBP2 deletion promotes hepatic gluconeogenesis and accelerates the progression of steatohepatitis. Conversely, activation of CtBP2 ameliorates diabetes and hepatic steatosis in obesity. The structure-function relationships revealed in this study identify a critical structural domain called Rossmann fold, a metabolite-sensing pocket, that is susceptible to metabolic liabilities and potentially targetable for developing therapeutic approaches.

[1] Department of Internal Medicine (Endocrinology and Metabolism), Faculty of Medicine, University of Tsukuba, 1-1-1 Tennodai, Tsukuba, Ibaraki 305-8575, Japan. [2] Transborder Medical Research Center, University of Tsukuba, 1-1-1 Tennodai, Tsukuba, Ibaraki 305-8577, Japan. [3] Department of Chemistry, Rikkyo University, Nishi-Ikebukuro, Toshima, Tokyo 171-8501, Japan. [4] Graduate Division of Nutritional and Environmental Sciences, University of Shizuoka, 52-1 Yada, Suruga-ku, Shizuoka 422-8526, Japan. [5] The Institute of Medical Science, The University of Tokyo, 4-6-1, Shirokanedai, Minato-ku, Tokyo 108-8639, Japan. [6] Department of Bioengineering, School of Engineering, The University of Tokyo, 7-3-1, Hongo, Bunkyo-ku, Tokyo 113-8656, Japan. ✉email: msekiya@md.tsukuba.ac.jp

Metabolism is one of the fundamental biological processes in all living cells and organisms to maintain their energy balance, structural components, and integrity. Recently, numerous metabolic intermediates generated throughout the complex metabolic networks have emerged as multifunctional regulators orchestrating cellular homeostasis: epigenetic regulation, modulation of enzymatic activities, posttranslational modification, and so forth[1,2]. This metabolite-driven system has been reported to influence a wide variety of biological processes and diseases. Aerobic glycolysis is a metabolic hallmark of cancer cells where the glycolytic metabolic intermediates evoke so-called metabolic reprogramming to further fuel cell growth and proliferation[3]. Cellular glycolysis and fatty acid oxidation determine the fate of polarization in immune cells[4] as well as in intestinal epithelial cells[5]. Thus, the molecular link between metabolism and cellular response programs is of growing interest in various fields.

Among the metabolic abnormalities associated with obesity, unsuppressed gluconeogenesis in tandem with increased lipogenesis is a key metabolic feature observed specifically in the liver of obesity[6]. Potentiation of the insulin signaling, which most of current therapeutic approaches for diabetes rely on, suppresses gluconeogenesis at the expense of induction of lipogenesis that potentially aggravates hepatosteatosis, indicating lack of appropriate pharmacological target(s) to properly suppress both of these metabolic pathways. This therapeutic impasse may exist in part because this complex hepatic biology has been investigated mostly from hormonal signaling perspectives while metabolite-centered approaches remain premature. Among metabolic signatures associated with obesity, significant contributions of dysregulation of fatty acid homeostasis have been well established[7–11]. Unsuppressed adipose lipolysis, known to significantly contribute to excessive hepatic glucose production[12,13], initiates this dysregulated fatty acid homeostasis by mobilizing fatty acids into circulation. Fatty acid influx into liver yields fatty acyl-CoAs as the first intermediates that have been shown to play multifaceted roles in physiological and pathological conditions[14,15]. The fatty acyl-CoAs are further metabolized to yield acetyl-CoA in mitochondria that activates pyruvate carboxylase to stimulate gluconeogenesis[16]. Considering the functional interdependence between fatty acid metabolism and NADH/NAD$^+$ regulation[17] and a role for NADH/NAD$^+$ ratio associated with mitochondrial activity in hepatic glucose production[18,19], pyridine dinucleotides may also be involved. These metabolic intermediates and their molecular targets would be of particular interest in understanding the dysregulation of hepatic glucose and lipid metabolism.

Through a structural pocket called Rossmann fold domain, the C-terminal binding proteins (CtBPs) can accommodate NADH/NAD$^+$ with preferential binding affinity for NADH and serve as redox-sensing transcriptional corepressors[20,21] albeit with some controversy regarding the discrimination between NADH and NAD$^+$[22–25]. There are two isoforms, CtBP1 and CtBP2, and only CtBP2 has a nuclear localization signal localized in its N-terminal region. This implicates the function of CtBP2 designated and specialized for transcription[26] while CtBP1, which lacks the nuclear localization signal, distributes throughout cells and non-transcriptional functions have also been attributed to CtBP1 (refs. [27,28]). The global deficiency of CtBP2 manifests embryonic lethality while that of CtBP1 exhibits milder phenotype due to the developmental defects in both models[29]. In part because of this lethality, investigations focused on metabolism or in vivo function of CtBPs have not been previously described.

Here we identified a metabolic system orchestrated by CtBP2 in response to key metabolic intermediates that controls hepatic glucose and lipid metabolism. Our findings illustrate the involvement of this system in the pathogenesis of obesity that is targetable to develop therapeutic approaches.

## Results

### Unraveling CtBP2 cistrome leads to an identification of CtBP2/FoxO1 complex as a regulatory node of hepatic gluconeogenesis.
Apart from the redox coupling, NADH functions as an intracellular energy carrier as well, and redox state is inseparably connected to cellular energy metabolism. Since our preliminary data obtained in the screening stage indicated a possible direct involvement of CtBP2 isoform rather than CtBP1 in hepatic glucose metabolism, we initiated our study to examine the potential biological roles of CtBP2 in the liver.

At first, we attempted to decipher the CtBP2 cistrome in normal liver tissues using chromatin immunoprecipitation (ChIP) followed by high-throughput DNA sequencing (ChIP-seq) since the global transcriptional landscape would provide clues to the molecular basis regulated by CtBP2 in an unbiased manner. The analysis of CtBP2-binding sites with transcriptional elements revealed that CtBP2 is more frequently recruited into transcriptional start site (TSS) although ChIP peaks are also observed on gene bodies of a certain set of genes (Supplementary Fig. 1a, b). Interestingly, NADH/NAD$^+$ metabolism, redox reactions, and epigenetic gene regulation were ranked on the top of the gene ontology analysis (Fig. 1a), which is consistent with the reported functions of CtBP2. In line with our hypothesis, one of the metabolic pathways, fatty acid biosynthetic pathway, was also ranked in the list. Indeed, CtBP2 was recruited to genomic regions encoding fatty acid biosynthesis genes (Fig. 1b and Supplementary Fig. 1c). Intriguingly, CtBP2 was also recruited into genomic loci encoding gluconeogenetic genes and inflammatory genes, critical for the pathogenesis of metabolic diseases (Fig. 1b and Supplementary Fig. 1c). Since CtBP2 lacks a DNA-binding domain, it is postulated that CtBP2 regulates transcription through binding to transcription factors bound to genomic regions[21], and a comprehensive understanding of its binding partners is critical to reveal the regulatory mechanisms of CtBP2. As a first step, we extracted consensus motifs from ChIP-seq peaks and found 25 motifs with e-values of less than 1.0e−04 (most of these motifs are shown as boxed sequence logos in Fig. 1c and Supplementary Fig. 1d). To find candidate transcription factors bound to CtBP2, we further compared these motifs against databases of known motifs. In agreement with the occupancy of CtBP2 in metabolic genes involved in hepatic gluconeogenesis and lipogenesis, the list of candidate transcription factors contained master regulators for both pathways, Forkhead box O1 (FoxO1)[30,31], sterol regulatory element-binding protein1 (SREBF1, SREBP1)[32], and MLX interacting protein-like (MLXIPL or carbohydrate responsive element-binding protein, ChREBP) (Fig. 1c and Supplementary Fig. 1d)[33] although none of these transcription factors have been reported as binding partners for CtBP2. The list of transcription factors contains other transcription factors previously reported to directly bind to CtBP2 such as KLF4 (ref. [34]) and E2F7 (ref. [35]) or indirect binding partners such as GATA family transcription factors[35] with higher e-values for direct binding partners, ensuring reliability of our analysis (Supplementary Fig. 1d). Intriguingly, we could find a putative CtBP interaction site, Pro-x-Asp-Leu motif[21], in the C-terminus of FoxO1 protein (Supplementary Fig. 2a, b) whereas the amino acid sequences of SREBP1 and ChREBP lack this CtBP-binding motif. Together with the e-values for these transcription factors, CtBP2 may bind to FoxO1 directly and to SREBP1 and/or ChREBP indirectly. Indeed, reciprocal co-immunoprecipitation experiments demonstrated the presence of an endogenous CtBPs/FoxO1 complex in primary hepatocytes (Supplementary Fig. 2c). Notably, mutation or deletion of the PSDL motif identified in FoxO1 specifically reduced the interaction between CtBP2 and FoxO1 but did not diminish interactions between CtBP1 and FoxO1 or another

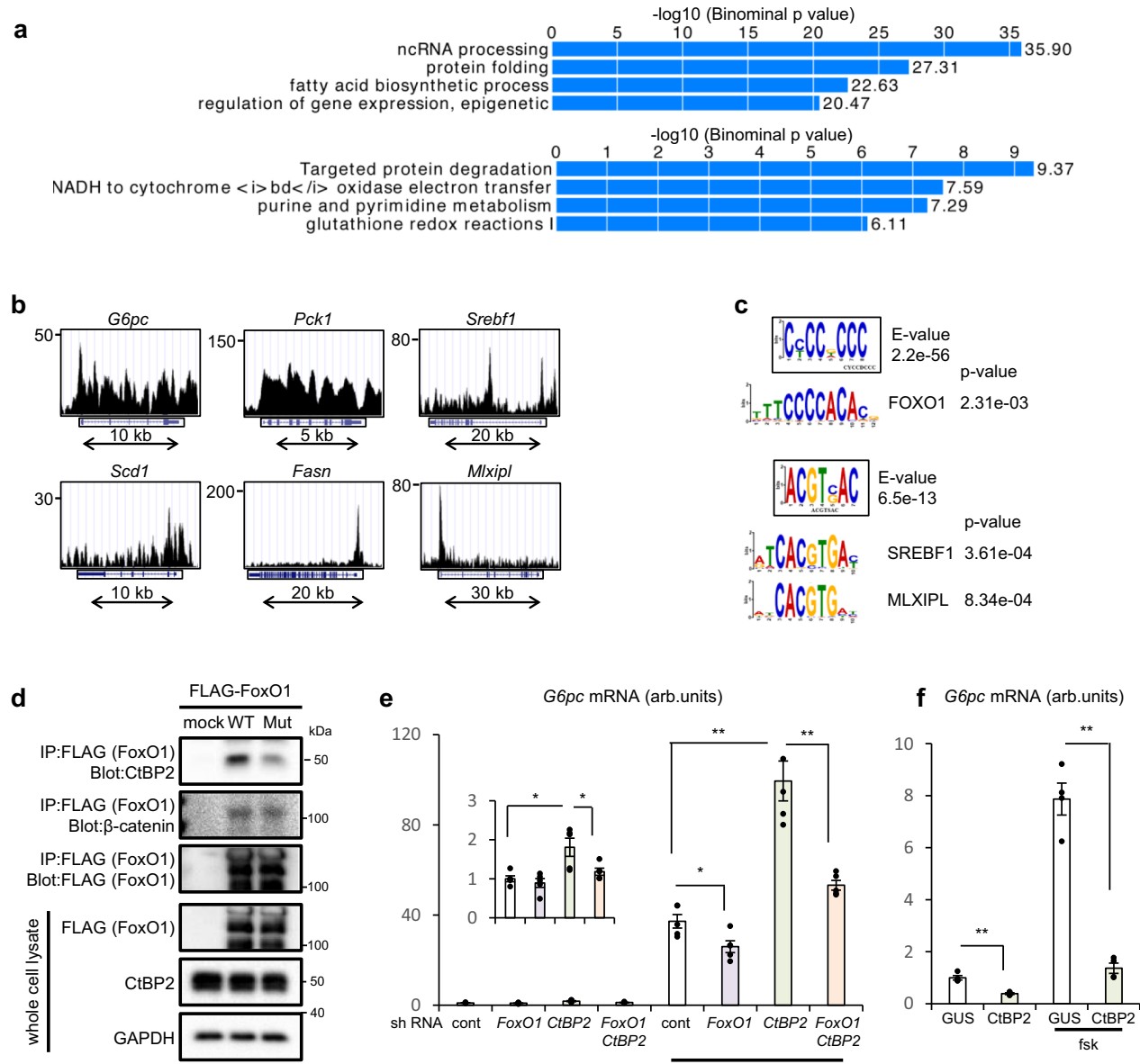

**Fig. 1 Unraveling CtBP2 cistrome leads to an identification of CtBP2/FoxO1 complex. a–c** CtBP2 ChIP-seq analysis in normal mouse liver tissues (6-h fasted). **a** Gene ontology analysis. Upper: biological process, lower: BioCyc pathway. **b** ChIP-seq peaks at representative metabolic gene loci. **c** Motif analysis of CtBP2 binding sites. The motifs enclosed by a rectangle are CtBP2-binding motifs enriched in the ChIP-seq. Prediction of known transcription factors targeted by CtBP2 based on sequence similarity is shown below. The statistical evaluation was performed based on the specific algorithm[85]. **d** Mutation of the PSDL motif in FoxO1(Mut: PSDL > PSAS) diminishes CtBP2/FoxO1 interaction in HEK293 cells. **e, f** Primary hepatocytes were stimulated with vehicle or 2 μM forskolin (fsk) after knockdown (**e**) or overexpression (**f**) of CtBP2. **e** The effect of CtBP2 and/or FoxO1 knockdown on *G6pc* expression ($n = 5$, biologically independent cells). The $y$-axis scale is expanded in the inset to show the data in the absence of fsk. The $p$ values are as follows: 0.011 for sh-cont vs sh-*CtBP2*, 0.036 for sh-*CtBP2* vs sh-*FoxO1*, 0.022 for sh-cont/fsk vs sh-*FoxO1*/fsk, $1.6 \times 10^{-4}$ for sh-cont/fsk vs sh-*CtBP2*/fsk, $9.4 \times 10^{-4}$ for sh-*CtBP2*/fsk vs sh-*FoxO1/CtBP2*/fsk. **f** *G6pc* expression following CtBP2 overexpression ($n = 4$, biologically independent cells). The $p$ values are as follows: $5.6 \times 10^{-4}$ for GUS vs CtBP2, $5.6 \times 10^{-5}$ for GUS/fsk vs CtBP2/fsk. Data are expressed as the mean ± SEM. * and ** denote $p < 0.05$ and $p < 0.01$ evaluated by unpaired two-tailed Student's $t$-test, respectively. Source Data are provided as a Source Data file.

FoxO1-binding partner, β-catenin, in HEK293 cells (Fig. 1d and Supplementary Fig. 2d, e), indicating that CtBP2 directly and specifically binds to FoxO1 through its PSDL motif whereas CtBP1 may bind to FoxO1 indirectly or through other interaction site(s).

We validated that the gluconeogenic transcriptional program was indeed regulated by the corepressor CtBP2. Suppression of *CtBP2* increased the expression of *G6pc* (Fig. 1e and Supplementary Fig. 2f, g) in primary hepatocytes, and this effect was enhanced in the presence of forskolin, an activator of cAMP-

generating systems. Furthermore, the induction of these genes following *CtBP2* suppression was prevented by simultaneous suppression of *FoxO1* (Fig. 1e). In line with these findings, *CtBP2* suppression resulted in activation of the Forkhead response element (FHRE) luciferase reporter (Supplementary Fig. 2h). Conversely, cells with exogenous CtBP2 expression (Supplementary Fig. 2i) showed reduced *G6pc* expression at baseline and robust suppression of the forskolin-induced increase in *G6pc* (Fig. 1f) mRNA levels compared to cells expressing a control gene (β-glucuronidase, GUS) as well as FHRE luciferase reporter

activity (Supplementary Fig. 2j). These data indicate that CtBP2 suppresses FoxO1 to repress gluconeogenic transcriptional programs.

**CtBP2 is a metabolite sensor responding to NADH and fatty acyl-CoAs.** Next, we investigated the previously reported redox-sensing function of CtBP2. Increasing concentrations of NADH promoted the formation of CtBP2/FoxO1 complex (Fig. 2a), proving evidence linking this gluconeogenic regulatory node to cellular redox state. Although CtBPs have not been previously investigated as a fatty acyl-CoA sensor, several lines of evidence prompted us to test this possibility: (1) our ChIP-seq data indicated CtBP2 could be tightly connected to fatty acid biosynthetic pathway, (2) the Rossmann fold proteins bind to adenosine[36], a structural component shared by NADH/NAD$^+$ and CoAs (Supplementary Fig. 3a), and (3) a structural study showed that CtBPs may interact with fatty acyl-CoAs[37]. Indeed, we were able to demonstrate the direct interaction between CtBP2 and oleoyl-CoA in vitro using microscale thermophoresis (MST)[38] where the dissociation constant $K_d$ of this interaction was $18.8 \pm 1.28 \,\mu M$ while that between CtBP2 and NADH was $19.5 \pm 6.61 \,\mu M$ (Fig. 2b). Our computer-assisted structural analysis further supported this hypothesis (Fig. 2c). As predicted, palmitoyl-CoA binds to CtBP2 having its CoA moiety in the Rossmann fold due to the strong electrostatic interactions. Interestingly, the acyl-chain moiety was computationally predicted to reside at the interface of CtBP2 dimer to block dimerization that is required for CtBP2 activation[39]. Indeed, unlike NADH, the presence of fatty acyl-CoA resulted in disruption of the CtBP2/FoxO1 interaction in a dose-dependent manner (Fig. 2d). In contrast to this, sodium oleate which lacks CoA moiety did not influence the CtBP2/FoxO1 interaction (Supplementary Fig. 3b) further supporting our hypothesis. As an alternative approach to validate the metabolite recognition by CtBP2, we performed the fluorescence-based thermal shift assay known as differential scanning fluorimetry (DSF)[40]. In this assay, the presence of NADH increased conformational stability of CtBP2 upon thermal denaturation while that of oleoyl-CoA decreased the melting temperature, indicating direct interaction of CtBP2 with both metabolic intermediates and opposing effects of these metabolic intermediates on the allosteric conformational transition of CtBP2 protein (Fig. 2e). We further attempted to obtain a rough estimate of $K_d$ with this assay by titrating ligand concentrations: $K_d$ for NADH was ~10.9 $\mu M$ while that for oleoyl-CoA was ~17.6 $\mu M$ (Supplementary Fig. 3c). Collectively, these data indicate that CtBP2 complex formation required for its co-repressor activity is stabilized by NADH binding and disrupted by fatty acyl-CoA binding.

Having observed these metabolite-sensing capabilities of CtBP2, we examined the effects of alteration of cellular metabolism on CtBP2/FoxO1 interaction. Since lactate dehydrogenase is an equilibrium enzyme coupling conversion of pyruvate to lactate with NADH to NAD$^+$, it is possible to modulate cytosolic NADH/NAD$^+$ ratio by changing extracellular lactate/pyruvate ratio[41]. Formation of the CtBP2/FoxO1 complex was enhanced in cells with a high extracellular lactate/pyruvate ratio (high cytosolic NADH/NAD$^+$ ratio) (Fig. 2f). Glycolysis is also coupled to conversion from NAD$^+$ to NADH through the enzymatic activity of GAPDH (glyceraldehyde-3-phosphate dehydrogenase). Koningic acid (KA), a specific GAPDH inhibitor, suppressed the CtBP2/FoxO1 complex formation, supporting NADH/NAD$^+$ sensing function of this complex (Supplementary Fig. 3d). In an additional experiment, we tested responsiveness to changes in fatty acyl-CoA concentrations. Increasing the cellular fatty acyl-CoA content by exogenously

supplied fatty acids with long-chain fatty acyl-CoA synthetase-1 (ACSL1) expression resulted in dissociation of the CtBP2/FoxO1 complex (Fig. 2g). We also monitored the CtBP2/FoxO1 complex formation in living cells over time using a split luciferase complementation assay (Fig. 2h). Stimulation of the cells stably expressing the reporter with lactate resulted in a rapid and robust increase of the signals even within a minute (Fig. 2i). This observation further supports our hypothesis that CtBP2 responds to metabolic alterations directly and quickly rather than to the cellular events induced indirectly.

**CtBP2 is inactivated in obesity.** Since the CtBP2/FoxO1 interaction can serve as a regulatory switch for hepatic gluconeogenesis responding to metabolic alterations, next we examined in vivo relevance of this transcriptional complex. Firstly, we examined the responsiveness of this complex to physiological conditions: a fasting feeding cycle, where fasting decreased the CtBP2/FoxO1 complex formation only marginally (Fig. 3a). Since there are numerous metabolic alterations occurring during this cycle, we next examined the responsiveness of this complex to representative metabolic cues. The CtBP2/FoxO1 complex was markedly diminished in response to the gluconeogenic hormones, glucagon and glucocorticoid (Fig. 3b and Supplementary Fig. 4a) while it was increased in response to insulin (Fig. 3c), consistent with its repressive role in hepatic gluconeogenesis. Unexpectedly, the CtBP2/FoxO1 complex formation was diminished by glucose administration (Fig. 3d). The CtBP2/FoxO1 complex formation in response to the fasting feeding cycle appears to be reflecting the net effects of these responses. Next, we investigated the responsiveness to pathological conditions using models of obesity. In mice with genetic obesity, CtBP2/FoxO1 interaction was dramatically (~90%) reduced (Fig. 3e) as well as in diet-induced obese mice (Fig. 3f). We also examined human liver specimens to examine whether this defect in CtBP2/FoxO1 interaction seen in multiple preclinical models is also present in human disease. Liver autopsy samples showed diminished CtBP2/FoxO1 interaction in the liver of subjects with obesity, suggesting that our findings could be extrapolatable and relevant to human disease despite the imperfect qualities of samples related to the postmortem changes (Fig. 3g). Taken together, our in vivo observations suggest that CtBP2 is inactivated in obese liver leading to FoxO1 liberation that could result in unsuppressed hepatic glucose production. We further examined a possible interaction of CtBP2 with SREBP1 or ChREBP although the interaction(s) may be indirect (Fig. 1c). Interestingly, CtBP2 formed an interaction with SREBP1 in the liver of lean mice which was diminished in both genetic obesity and diet-induced obesity (Fig. 3h, i). On the other hand, we could not detect an interaction between CtBP2 and ChREBP in a convincing manner, which might be reflecting some technical limitations or indirect nature of binding. To ascertain possible repressing effects of CtBP2 on lipogenic genes, we examined the expression of lipogenic genes in the same experimental setting as described in Fig. 1f. Indeed, the expression of lipogenic genes was repressed by exogenous expression of CtBP2 in primary hepatocytes (Fig. 3j). To further characterize the CtBP2/SREBP1 interaction, we attempted to find the intermediary molecule between these molecules. Among the reported SREBP1 partners, we found the putative CtBP-binding motif in NR5A2 (nuclear receptor subfamily 5 group A member 2 or LRH-1, liver receptor homolog-1) that was reported to have a repressive effect on SREBP1 (ref. [42]) (Supplementary Fig. 4b). As expected, we observed an interaction between CtBP2 and NR5A2(LRH-1), and mutation or deletion of the CtBP-binding motif in NR5A2(LRH-1) diminished the interaction (Fig. 3k and Supplementary Fig. 4c). Exogenous expression of these three components

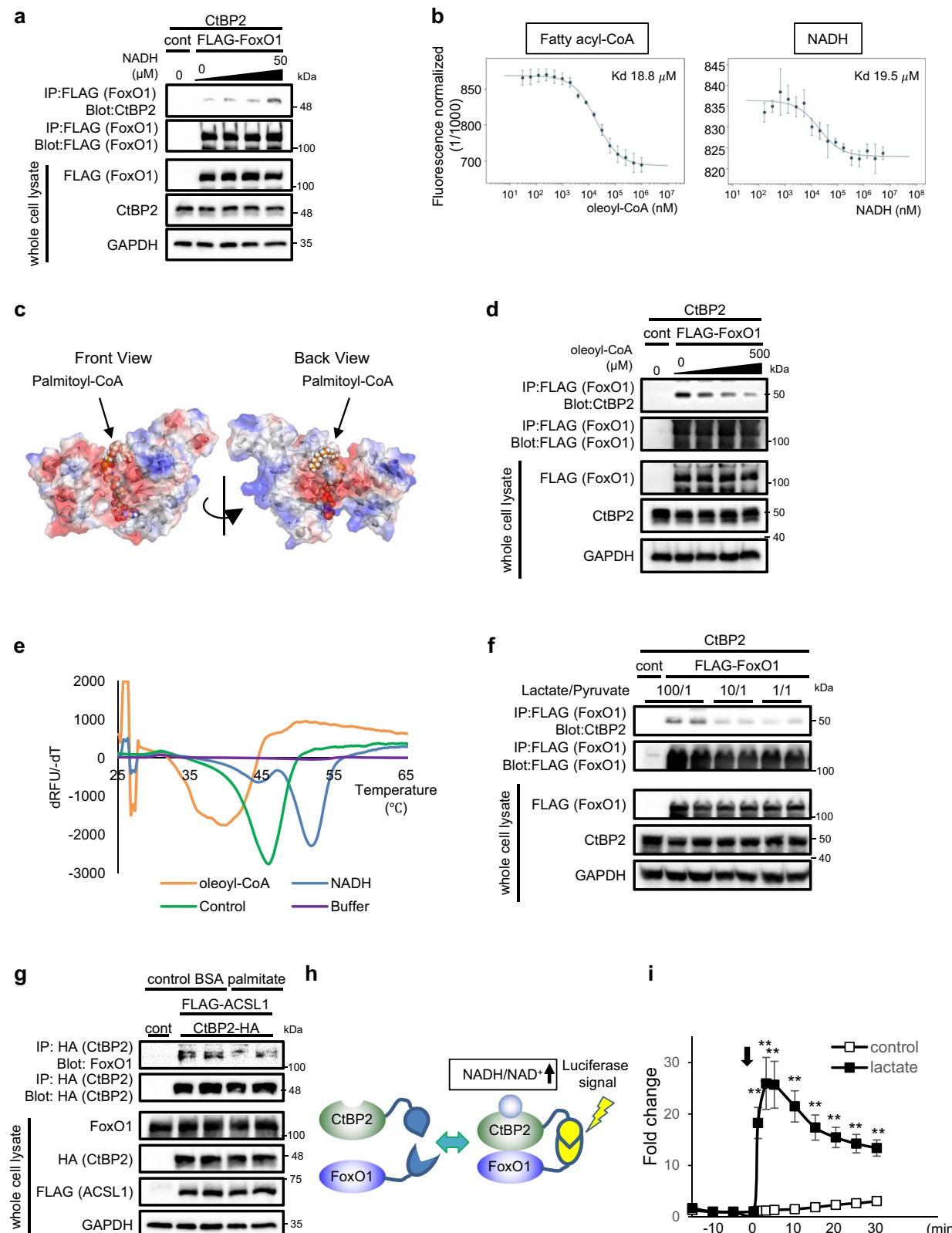

reconstituted the CtBP2/NR5A2(LRH-1)/SREBP1 complex in vitro (Supplementary Fig. 4d). Moreover, suppression of *NR5A2(LRH-1)* in normal liver tissues decreased the CtBP2/SREBP1 interaction in vivo (Supplementary Fig. 4e and Fig. 3l). These data provide strong support for our hypothesis that CtBP2 interacts with SREBP1 through NR5A2(LRH-1). We further examined whether the diminished binding of CtBP2 observed in

obesity is specific to FoxO1 and SREBP1 by detecting the interaction of CtBP2 with a known partner, zinc-finger protein FOG family member 1 (ZFPM1), that was again decreased in the liver of obesity (~96%, Supplementary Fig. 4f). These findings led us to examine the levels of metabolic intermediates controlling CtBP2 activity in vivo. Nuclear fatty acyl-CoA content was increased in the liver of two models of obesity (Fig. 3m, n) consistent with a

**Fig. 2 CtBP2 confers metabolite-sensing capabilities to FoxO1. a** The NADH-sensing capability of CtBP2/FoxO1 complex. Increasing concentrations of NADH (0, 0.5, 5, 50 μM) were added to the HEK293 cell lysates expressing indicated proteins. **b** Direct binding of recombinant CtBP2 protein to oleoyl-CoA and NADH analyzed by MST ($n = 3$ or 4 for each point). **c** Computer-assisted structural analysis of CtBP2 with palmitoyl-CoA. Palmitoyl-CoA is localized at the interface of CtBP2 adopting dimeric configuration. **d** The fatty acyl-CoA-sensing capability of CtBP2/FoxO1 complex. Increasing concentrations of oleoyl-CoA (0, 50, 150, 500 μM) were added to the HEK293 lysates. **e** The thermal shift induced by NADH and oleoyl-CoA binding in DSF. Recombinant CtBP2 were mixed with either NADH (20 μM) or oleoyl-CoA (20 μM) and thermal stability was monitored on a temperature gradient. **f** The effect of cytosolic redox state on CtBP2/FoxO1 complex. HEK293 cells expressing FLAG-FoxO1 and CtBP2 were incubated with different ratios of lactate and pyruvate for 1 h. **g** HEK293 cells expressing indicated proteins were incubated with control BSA or palmitate-BSA conjugate (200 μM) for 30 min. **h** Schematic diagram showing the split luciferase complementation assay. **i** HEK293 cells stably expressing the reporter were stimulated with 50 mM lactate. The arrow indicates the addition of either vehicle or lactate ($n = 4$, biologically independent cells). The $p$ values are as follows: $1.3 \times 10^{-3}$ for 1 min, $2.8 \times 10^{-3}$ for 3 min, $1.8 \times 10^{-3}$ for 5 min, $6.0 \times 10^{-4}$ for 10 min, $7.3 \times 10^{-4}$ for 15 min, $5.6 \times 10^{-4}$ for 20 min, $6.5 \times 10^{-4}$ for 25 min, $6.2 \times 10^{-4}$ for 30 min. Data are expressed as the mean ± SEM. ** denotes $p < 0.01$ evaluated by unpaired two-tailed Student's $t$-test. Source Data are provided as a Source Data file.

previous report[43], implicating increased fatty acyl-CoAs could cause the dissociation of CtBP2/FoxO1 complex observed in the liver of obesity. Hepatic lactate/pyruvate ratio reflecting cytosolic NADH/NAD$^+$ ratio was also decreased in the liver of diet-induced but not in the genetically obese mice (Supplementary Fig. 4g, h), implicating a predominant role of fatty acyl-CoA in the regulation of CtBP2 activity compared to NADH/NAD$^+$ ratio. In agreement with these observations, supplementation of oleoyl-CoA in liver lysates from normal mice reduced CtBP2/SREBP1 complex formation (Supplementary Fig. 4i), implicating fatty acyl-CoA-mediated CtBP2 inactivation in obesity would lead to the concurrent upregulation of hepatic gluconeogenesis and lipogenesis. On the other hand, these two pathways are reciprocally regulated in physiology. Therefore, we further examined the CtBP2/SREBP1 complex formation in physiological conditions. The CtBP2/SREBP1 complex formation was increased by glucagon and marginally decreased by glucocorticoid, consistent with the effects of these stimuli on hepatic lipogenesis (Supplementary Fig. 4j). In contrast, fasting feeding cycle and other stimuli did not change the CtBP2/SREBP1 interaction (Supplementary Fig. 4j). The regulation of CtBP2/SREBP1 complex in physiology would be more complicated by the presence of the intermediary molecule or other unknown systems.

**Inactivation of CtBP2 contributes to the pathogenesis of obesity-related metabolic disturbances**. Observing the dissociation of CtBP2 from FoxO1 in obesity, a question arises regarding in which cellular compartment this regulatory step occurs. To address this question, we quantified the recruitment of CtBP2 to the promoters of gluconeogenic genes by ChIP. The recruitment of CtBP2 to gluconeogenic gene promoters was decreased in the liver of obese mice (Fig. 4a and Supplementary Fig. 5a). Together with the fact that CtBP2 localizes exclusively in the nuclei[26], CtBP2/FoxO1 complex may exist on the DNA elements in the nuclei and CtBP2 may dissociate from the FoxO1 bound to those elements. Supporting this idea, re-ChIP experiments demonstrated that CtBP2 and FoxO1 shared occupancy of the *G6pc* promoter in unstimulated cells (Fig. 4b) and FoxO1 could be detected on that promoter even in the lean animals (Fig. 4c and Supplementary Fig. 5b). FoxO1 protein was present in isolated nuclei irrespective of the feeding conditions or the degree of obesity although 24-h fasting modestly increased the nuclear FoxO1 (Supplementary Fig. 5c). In our subcellular localization study, exogenously expressed CtBP2 localized exclusively in the nuclei and did not alter subcellular trafficking of FoxO1 that was affected by insulin signaling as reported previously (Fig. 4d). To further validate the CtBP2/FoxO1 interaction in the nuclei, we took advantage of a FoxO1-mutant lacking the DNA-binding domain (△DB). The mutant FoxO1 was forced to be localized mostly in the cytoplasm as reported (Supplementary Fig. 5d)[44].

The presence of the mutation diminished the CtBP2/FoxO1 interaction showing another layer of evidence indicating this interaction occurs in the nuclei (Fig. 4e). Since the DNA-binding domain in FoxO1 is apart from the CtBP-binding site (Supplementary Fig. 2b), the △DB mutation itself would not affect CtBP2/FoxO1 interaction. Indeed, purified recombinant △DB FoxO1 mutant interacted with recombinant CtBP2 protein in vitro (Supplementary Fig. 5e), suggesting the diminished interaction between CtBP2 and the FoxO1 △DB mutant can be explained by the cellular compartmentalization.

The observation of robust inactivation of CtBP2 in obesity prompted us to utilize a liver-specific CtBP2-deletion model in mice. For this, we generated CtBP2 flox mice (Supplementary Fig. 6a, b) and crossed them with the albumin-Cre transgenic mice. We confirmed liver-specific CtBP2-knockout at the protein level with some residual expression while CtBP1 expression was not altered upon removal of CtBP2 (Fig. 5a). There was no discernible change in body weights (Supplementary Fig. 6c). However, these animals did exhibit impaired glucose tolerance (Fig. 5b) where no difference was evident in insulin sensitivity (Supplementary Fig. 6d). As predicted, hepatic deletion of CtBP2 led to increased expression of gluconeogenic genes (Fig. 5c), suggesting that hepatic glucose production may be the primary target of CtBP2. We further evaluated hepatic gluconeogenesis by pyruvate tolerance test and demonstrated a significant increase in glycemia following pyruvate administration in the liver-specific CtBP2-deficient mice (Fig. 5d). While plasma insulin levels were not influenced by the CtBP2-deficiency, plasma glucagon levels were decreased (Supplementary Fig. 6e,f), which might be explained as a compensatory decrease in response to the elevated hepatic glucose output. It is also of note that plasma insulin levels were unchanged even in the presence of elevated glucose and decreased glucagon, that may also suggest existence of some systemic effects influencing insulin biology. In addition to the impairment of glucose tolerance, hepatic deletion of CtBP2 also caused a modest increase in liver triglyceride content in animals maintained on a regular chow diet (Fig. 5e) along with a trend of an increase in the expression of lipogenic genes (Fig. 5f). The expression of some hepatocyte genes such as albumin (*Alb*) was not influenced (Supplementary Fig. 6g). In addition, we analyzed plasma lipid profile considering potential contribution of lipid fluxes to liver triglyceride content. Plasma total cholesterol was marginally increased in CtBP2-deficiency while plasma triglyceride was unchanged (Supplementary Fig. 6h). We also examined the expression of FoxO1 and SREBP1 in this mouse model at the protein levels. Unexpectedly, expression levels of both FoxO1 and SREBP1 were decreased in CtBP2-deficiency, suggesting the existence of another compensatory mechanism (Fig. 5g and Supplementary Fig. 6i). Despite the increased hepatic triglyceride content, the liver-specific CtBP2-deficient mice did not show any

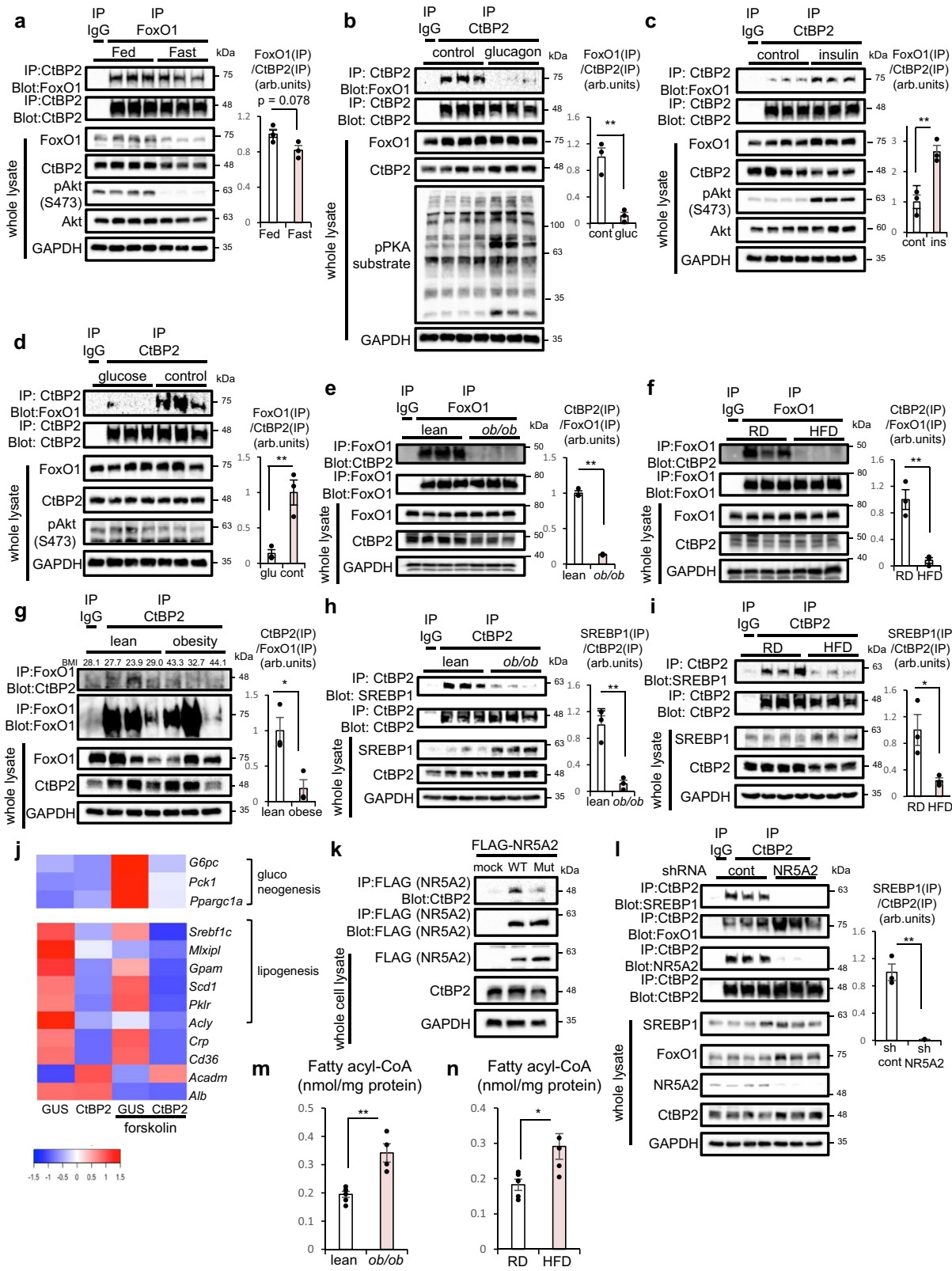

signs of liver dysfunction on a regular chow diet (Fig. 5h). To examine the involvement of CtBP2 in liver dysfunction in steatohepatitis, we fed mice a methionine choline-deficient (MCD) diet, a commonly used model to generate non-alcoholic steatohepatitis (NASH) in rodents. Interestingly, even after 1 week of the dietary challenge, the liver-specific CtBP2-deficient mice

exhibited hepatic steatosis (Fig. 5i and Supplementary Fig. 6j) with liver dysfunction as indicated by the elevated serum alanine aminotransferase (ALT) levels (Fig. 5h). In this experimental setting, plasma triglyceride was slightly but significantly decreased, which may in part contributed to the aggravation of hepatic steatosis (Supplementary Fig. 6k).

**Fig. 3 CtBP2 is markedly inactivated in the liver of obesity.** Liver homogenates from mice: **a–d**, **l** normal mice, **e**, **h** genetically obese mice, **f**, **i** diet-induced obese mice (12–16 weeks on high-fat diet(HFD), RD: regular diet) and human liver autopsy specimens (**g**) were subjected to co-immunoprecipitation to examine endogenous CtBP2/FoxO1 complex (**a–g**) or CtBP2/SREBP1 complex (**h**, **i**, **l**, $n = 3$ or 4 biological independent animals as shown in the blots for each experimental group). The densitometric quantification is shown to the right of each blot. **a–d** CtBP2/FoxO1 complex in normal mice in response to either a fasting-feeding cycle (**a** 24-h-fasted or fed ad libitum), glucagon (**b** 30 min after injection, 200 μg/kg, 3-h food withdrawal, $p = 1.1 \times 10^{-3}$), insulin (**c** 30 min after injection, 0.4 U/kg, 3-h food withdrawal, $p = 6.2 \times 10^{-3}$), or glucose (**d** 30 min after injection, 2 g/kg, overnight fasted, $p = 9.5 \times 10^{-3}$). **e–i** CtBP2/FoxO1 complex in mouse models of obesity (**e**, **f** $p = 1.3 \times 10^{-5}$ and $4.1 \times 10^{-4}$, respectively) or in human obesity (liver autopsy samples. Body mass index (BMI) for each subject was indicated above the blot, **g** $p = 0.022$), or CtBP2/SREBP1 complex in mouse models of obesity (**h**, **i** $p = 3.9 \times 10^{-3}$ and 0.033, respectively). Mice were fasted for 4–6 h. **j** Primary hepatocytes were treated as in Fig. 1f and gene expression profile was analyzed. **k** Mutation of the PSDL motif in NR5A2(LRH-1) (Mut: PSDL > PSAS) diminishes CtBP2/NR5A2(LRH-1) interaction in HEK293 cells. **l** CtBP2/SREBP1 complex in normal liver tissues was diminished by *NR5A2(LRH-1)* knockdown in vivo (6-h fasted, $p = 1.2 \times 10^{-3}$). **m**, **n** Fatty acyl-CoA content in the nuclei isolated from lean control mice or genetically (ob, **m** $p = 9.3 \times 10^{-4}$)/diet-induced obese mice (HFD, **n** $p = 0.026$) ($n = 6$ for lean, $n = 4$ for ob, $n = 5$ for RD and HFD, biologically independent animals, 6-h fasted). Data are expressed as the mean ± SEM. * and ** denote $p < 0.05$ and $p < 0.01$ evaluated by unpaired two-tailed Student's *t*-test, respectively. Source Data are provided as a Source Data file.

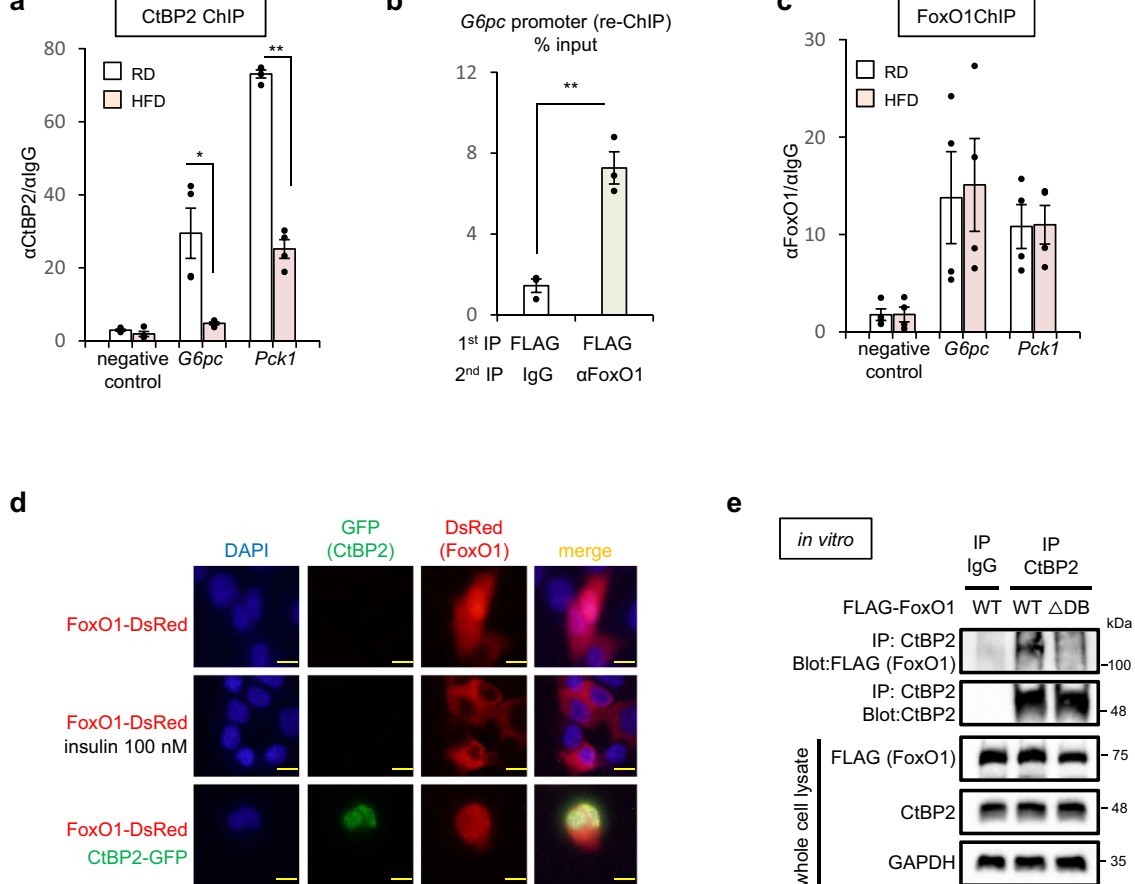

**Fig. 4 Dissociation of CtBP2 from FoxO1 occurs in the nucleus. a** CtBP2 recruitment to promoter regions of gluconeogenic genes ($n = 4$, biologically independent animals, *p* values are as follows: 0.011 for *G6pc* and $2.4 \times 10^{-6}$ for *Pck1*). HFD diet-induced obese mice. **b** Sequential ChIP (re-ChIP) in primary hepatocytes. FLAG-CtBP2 and FoxO1 were exogenously expressed, the chromatin was first enriched by FLAG (CtBP2) immunoprecipitation and the eluents were further immunoprecipitated with either control IgG or anti-FoxO1 antibody ($n = 3$, biologically independent cells, $p = 2.5 \times 10^{-3}$). **c** FoxO1 recruitment to promoter regions of gluconeogenic genes ($n = 4$, biologically independent animals). Liver tissues were collected following a 6-h fast (**a**, **c**). **d** FoxO1-DsRed and/or CtBP2-GFP were expressed in hepa1-6 cells and treated with vehicle or 100 nM insulin for 30 min. The yellow bars indicate 10 μm. **e** The forced cytoplasmic localization of FoxO1 markedly diminishes CtBP2/FoxO1 interaction. Either the wild-type FoxO1 (WT) or the mutant FoxO1 (△DB) was exogenously expressed in HEK293 cells, and the CtBP2/FoxO1 complex was co-immunoprecipitated. Data are expressed as the mean ± SEM. * and ** denote $p < 0.05$ and $p < 0.01$ evaluated by unpaired two-tailed Student's *t*-test, respectively. Source Data are provided as a Source Data file.

**Therapeutic potential of CtBP2 activation in the liver of obese mice.** Since CtBP2 activation in hepatocytes downregulated gluconeogenic and lipogenic gene expression in vitro (Figs. 1f and 3j) and the activity of CtBP2 is diminished in obesity (Fig. 3e–i and Supplementary Fig. 4f), we investigated whether exogenous supplementation of CtBP2 can improve the metabolic profile of

obese mice in vivo (Supplementary Fig. 7a). Replenishment of CtBP2 normalized fasting blood glucose levels in obese mice without inducing any changes in body weights (Fig. 6a and Supplementary Fig. 7b) and improved glucose tolerance (Fig. 6b) although the degree of obesity was modest. To more directly assess hepatic gluconeogenesis, we performed a pyruvate

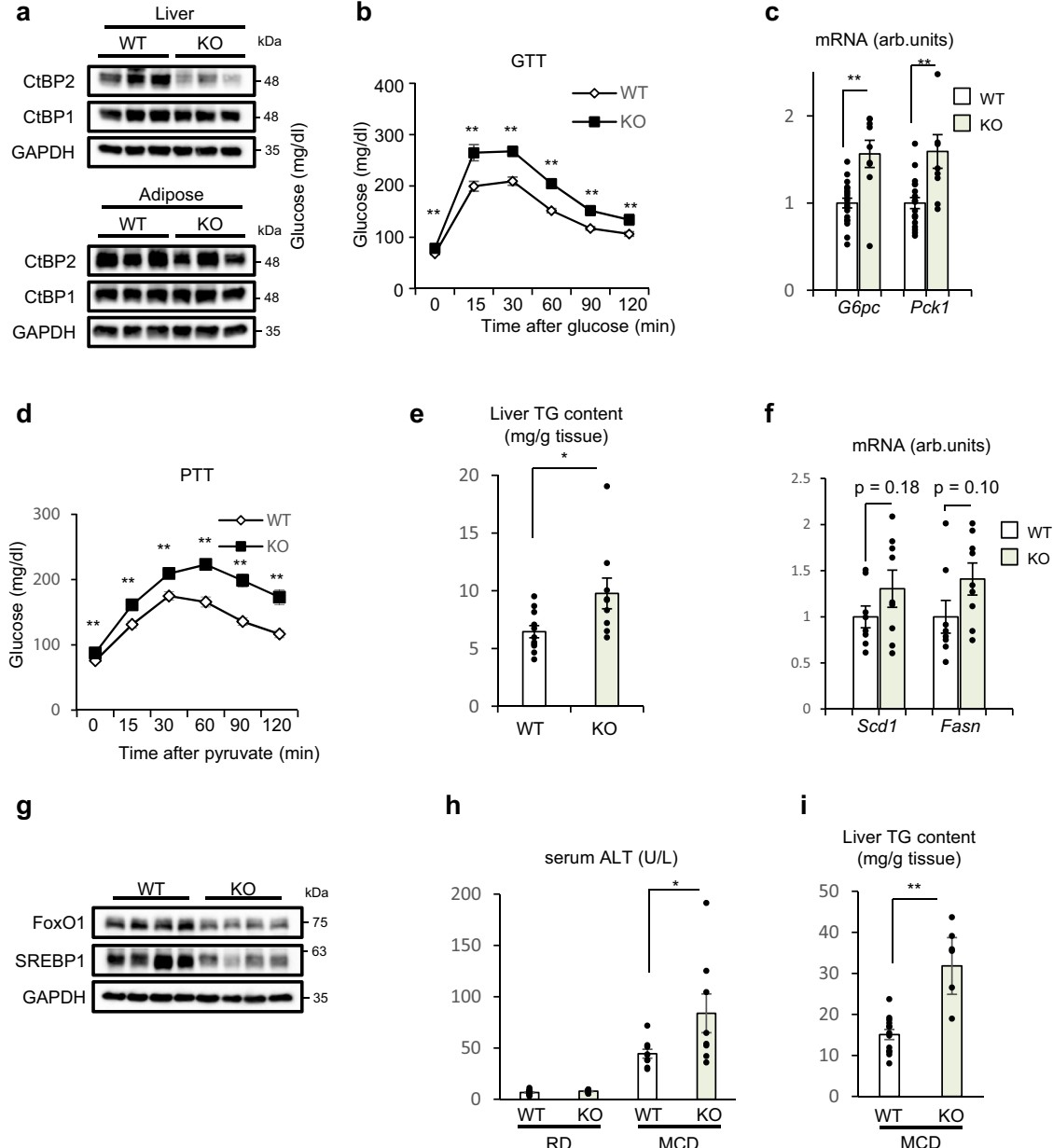

**Fig. 5 CtBP2 deficiency in liver leads to obesity-associated metabolic disturbances. a** Liver-specific CtBP2-deficient mice (KO) maintained on a regular chow diet (**a–h**) or an MCD diet for a week (**h**, **i**) (6–8 weeks of age, **a**, **c**, **e–i**: 6-h fasted, **b**, **d**: overnight fasted) were analyzed along with their wild-type controls (WT). **a** CtBP1 and CtBP2 expression at protein levels in either liver or adipose tissues. **b** Glucose tolerance test (GTT) ($n = 16$ for WT and $n = 8$ for KO, biologically independent animals, $p$ values are as follows: $4.3 \times 10^{-3}$ for 0 min, $4.4 \times 10^{-4}$ for 15 min, $2.2 \times 10^{-4}$ for 30 min, $2.0 \times 10^{-6}$ for 60 min, $2.6 \times 10^{-5}$ for 90 min, $8.1 \times 10^{-3}$ for 120 min). **c** Gluconeogenic gene expression ($n = 19$ for WT and $n = 9$ for KO, biologically independent animals, $p = 2.5 \times 10^{-4}$ and $1.0 \times 10^{-3}$ for *G6pc* and *Pck1*, respectively). **d** Pyruvate tolerance test (PTT) ($n = 12$ for WT and $n = 6$ for KO, biologically independent animals, $p$ values are as follows: $1.2 \times 10^{-3}$ for 0 min, $1.9 \times 10^{-3}$ for 15 min, $3.1 \times 10^{-3}$ for 30 min, $3.5 \times 10^{-4}$ for 60 min, $1.9 \times 10^{-5}$ for 90 min, $1.4 \times 10^{-4}$ for 120 min). **e** Liver triglyceride (TG) content ($n = 11$ for WT and $n = 9$ for KO, biologically independent animals, $p = 0.023$). **f** Lipogenic gene expression ($n = 8$ for WT and $n = 9$ for KO, biologically independent animals). **g** Expression levels of FoxO1 and SREBP1 at protein levels. **h** Serum ALT levels in mice fed either a regular chow diet (RD) or an MCD diet for a week ($n = 14$ for WT/RD, $n = 10$ for KO/RD, $n = 9$ for WT/MCD, $n = 8$ for KO/MCD, biologically independent animals, $p = 0.048$ for WT vs KO on an MCD diet). **i** Liver triglyceride (TG) content in mice fed an MCD diet for a week ($n = 13$ for WT and $n = 8$ for KO, biologically independent animals, $p = 8.3 \times 10^{-5}$). Data are expressed as the mean ± SEM. * and ** denote $p < 0.05$ and $p < 0.01$ evaluated by unpaired two-tailed Student's $t$-test, respectively. Source Data are provided as a Source Data file.

tolerance test, and observed that exogenous CtBP2 expression resulted in a significantly blunted glucose excursion following pyruvate injection (Fig. 6c). This improvement was not due to changes in insulin sensitivity, as despite differences in fasting glucose, there was no difference in the insulin tolerance when plotted as "percent of initial value" (Supplementary Fig. 7c, d).

Consistent with these findings, *G6pc* expression in liver was downregulated in CtBP2-overexpressing animals whereas expression of a non-target gene, *Alb*, was not affected (Fig. 6d). In agreement with these observations, the exogenous expression of CtBP2 restored the CtBP2/FoxO1 interaction (Supplementary Fig. 7e). CtBP2 has been reported to regulate gene expression

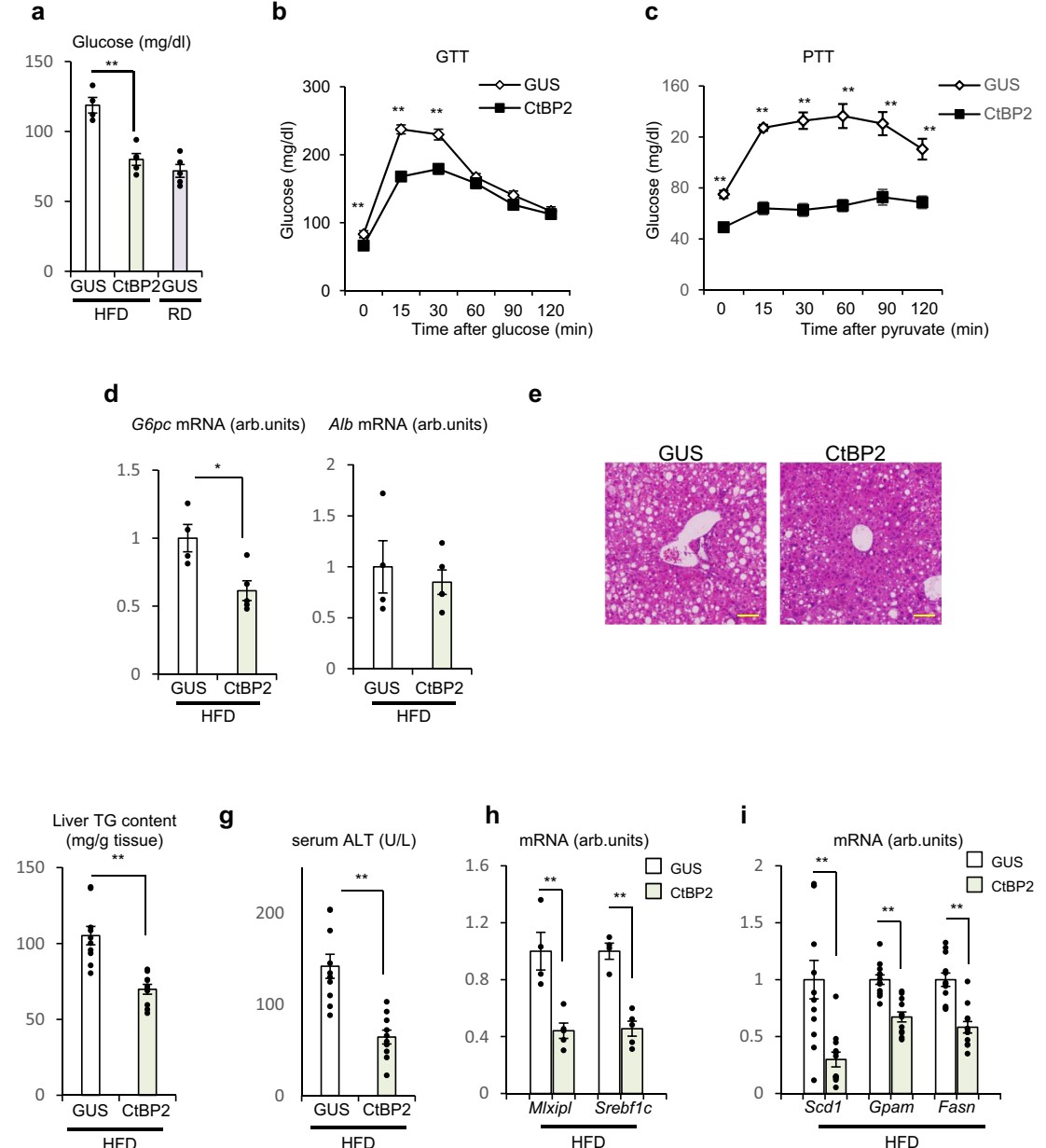

**Fig. 6 CtBP2 gain of function in liver ameliorates diabetes and hepatic steatosis in obese mice.** CtBP2 was exogenously expressed in the liver of diet-induced obese mice by adenoviral transduction (AdCtBP2), along with its control (AdGUS). Animals were analyzed after 3–7 days of transduction (**a**–**d, h** overnight fasted, **e**–**g, i** 6-h fasted). **a** Blood glucose levels ($n = 4$ for GUS/HFD, $n = 5$ CtBP2/HFD, and $n = 5$ for GUS/lean, biologically independent animals, $p = 7.8 \times 10^{-4}$). HFD high-fat diet, RD regular diet. **b** Glucose tolerance test (GTT, $n = 10$, biologically independent animals, $p$-values are $8.8 \times 10^{-3}$ for 0 min, $1.0 \times 10^{-8}$ for 15 min, $1.4 \times 10^{-4}$ for 30 min). **c** Pyruvate tolerance test (PTT, $n = 9$ for GUS, $n = 10$ for CtBP2, biologically independent animals, $p$ values are $7.6 \times 10^{-6}$ for 0 min, $2.5 \times 10^{-9}$ for 15 min, $1.1 \times 10^{-7}$ for 30 min, $2.6 \times 10^{-6}$ for 60 min, $4.9 \times 10^{-5}$ for 90 min, $3.0 \times 10^{-4}$ for 120 min). **d** Gene expression in liver ($n = 4$ for GUS, $n = 5$ for CtBP2, biologically independent animals, $p = 0.015$). **e** Representative hematoxylin and eosin stained sections of liver. Yellow bars indicate 100 μm. **f** Liver triglyceride (TG) content ($n = 10$, biologically independent animals, $p = 6.6 \times 10^{-5}$). **g** Serum ALT levels ($n = 10$, biologically independent animals, $p = 7.4 \times 10^{-5}$). **h, i** Lipogenic gene expression ($n = 4$ (**h**), $n = 10$ (**i**), biologically independent animals, $p = 3.8 \times 10^{-3}$ (*Mlxipl*) and $2.1 \times 10^{-4}$ (*Srebf1c*), $8.3 \times 10^{-4}$ (*Scd1*), $1.7 \times 10^{-5}$ (*Gpam*) and $2.0 \times 10^{-5}$ (*Fasn*), respectively). Data are expressed as the mean ± SEM. * and ** denote $p < 0.05$ and $p < 0.01$ evaluated by unpaired two-tailed Student's $t$-test, respectively. Source Data are provided as a Source Data file.

through alteration of epigenetic codes[21,45], and indeed, histone modifications associated with open chromatin at *G6pc* gene promoter were reduced by CtBP2 overexpression (Supplementary Fig. 7f). Furthermore, the exogenous expression of CtBP2 reduced hepatic lipid accumulation in obese mice (Fig. 6e and Supplementary Fig. 7g), liver triglyceride content (Fig. 6f), and serum ALT levels (Fig. 6g). Consistent with these findings, lipogenic

gene expression was also suppressed by CtBP2 overexpression (Fig. 6h, i).

**CtBP2 is a bona fide metabolite sensor for adenosine derivatives interconnecting cellular metabolic state to transcriptional programs.** Having observed the therapeutic potentials of CtBP2

activation in obesity, we decided to further explore the structure–function relationships in CtBP2 protein. For this purpose, we generated a mutant CtBP2 (mut) lacking the metabolite-sensing pocket Rossmann fold[39], and exogenously expressed it in the liver of obese mice along with wild-type CtBP2 (WT) and a control protein, GUS (Fig. 7a and Supplementary Fig. 8a). Intriguingly, while wild-type CtBP2 expression downregulated gluconeogenic and lipogenic genes as expected, the ability of CtBP2 to regulate these transcriptional programs was abolished by mutating Rossmann fold (Fig. 7a). We further obtained unbiased global expression profile of the livers of these mice using RNA sequencing (RNA-seq) to identify the genes regulated by CtBP2 in a Rossmann fold-dependent manner (Fig. 7b and Supplementary Data 1, $p$ value <0.05 for both GUS vs WT and WT vs mut, fold difference >1.25). The list comprises numerous metabolic genes, and the gene ontology analysis showed enrichment of genes associated with fatty acid metabolism, redox regulation, and glucose metabolism, suggesting that CtBP2 is a bona fide master regulator of global metabolism (Fig. 7c–e). Principal component analysis also supports the Rossmann fold-dependent transcriptional regulation (Fig. 7f). Moreover, KEGG pathway analysis showed most of the steps in fatty acid biosynthesis are under the control of CtBP2 in a Rossmann fold-dependent manner (Supplementary Fig. 8b). To better understand the role of the Rossmann fold, we performed a series of additional experiments. NADH-induced CtBP2/FoxO1 complex formation was competed with increasing concentrations of fatty acyl-CoA using cell lysates (Supplementary Fig. 8c). This observation was further confirmed using purified recombinant proteins to observe the specific interaction, where the metabolite sensing capability was abolished by the Rossmann fold mutation (Fig. 7g). The adenosine structure of NADH and palmitoyl-CoA shares the occupancy in the Rossmann fold in our structural modeling (Figs. 2c and 7h), which could explain the competitive nature of binding of these two metabolites. As we described, the acyl-chain moiety of palmitoyl-CoA located at the dimerization interface of CtBP2 to further inhibit CtBP2 activation (Figs. 2c and 7h). These observations raise a question whether, or to what extent, short chain acyl-CoAs would have inhibitory effects on CtBP2. Therefore, we additionally examined acetyl-CoA which has only two carbons in its acyl-chain with multifaceted roles in transcriptional regulation[46]. Intriguingly, acetyl-CoA decreased the CtBP2/FoxO1 complex formation in both experimental settings using cell lysates and recombinant proteins (Supplementary Fig. 8d and Fig. 7i) and the effect was modest compared to oleoyl-CoA. This modest effect of acetyl-CoA fits well with our model: acetyl-CoA can compete for the Rossmann fold with its CoA moiety while the short acyl-chain would be insufficient to physically block the dimerization. Our computer-assisted structural modeling further supported this idea (Supplementary Fig. 8e). The short acyl-chain protrudes from a CtBP2 molecule to prevent it from interacting with another CtBP2 molecule, which appears to be less potent to inhibit the dimerization compared to the long acyl-chain. Quantification of the interactions of recombinant proteins in Fig. 7g, i further clarified the effects of the acyl-chain length, where higher concentrations of acetyl-CoA were required to completely suppress the NADH-mediated formation of the CtBP2/FoxO1 interaction. We attempted to obtain more detailed structural information of CtBP2 complexed with palmitoyl-CoA by the first-principles calculations based on fragment molecular orbital (FMO) method. The residues which located close to the acyl-chain moiety were represented in Fig. 7j, suggesting that the acyl-chain moiety formed stable hydrophobic interactions. Especially, W324 appeared to have the highest interaction energy among those key residues (Supplementary Fig. 8g). The acyl-chain moiety located at the interface of CtBP2 dimer, which

ensures the reliability of our docking simulation (Figs. 2c and 7h) and further support our proposed model. To validate this in silico analysis, we generated alanine mutants for those key residues. Expression levels of these mutants were reduced compared to wild-type CtBP2 and interaction of those CtBP2 mutants with FoxO1 was markedly diminished, implicating these residues are critical for the dimeric activation of CtBP2 (Supplementary Fig. 8h). Although the diminished CtBP2/FoxO1 interaction hampered a fair assessment of the effects of those mutations on the responses to fatty acyl-CoA, H321A and W324A mutations appeared to abolish the responses to fatty acyl-CoA, that fits well with our interaction energy estimation (Supplementary Fig. 8g). Overall, acyl-chain moiety of fatty acyl-CoAs appears to interact with key residues in CtBP2 required for its dimeric assembly. Lastly, we performed molecular dynamics (MD) simulation where palmitoyl-CoA was allowed to interact with either monomeric form or dimeric form of CtBP2 (Supplementary Movies 1 and 2). MD simulation is a method for predicting the physical movements of biopolymers according to the Newtonian equation of motion. And this method can be applied to predict the thermodynamic stability of proteins-compounds complex. In this approach, the CoA moiety was accommodated in the Rossmann fold pocket in both forms of CtBP2 as expected. On the other hand, the acyl-chain moiety exhibited unrestricted movement with the monomeric form of CtBP2 while it was stabilized at the dimerization interface with dimeric form of CtBP2 where the fluctuation of the residues was quantified using root mean square fluctuations (RMSF) (Supplementary Fig. 8i). This additional study further supported our idea that the CoA moiety and the fatty acyl-chain moiety interact with the Rossmann fold and the dimerization interface, respectively. Moreover, these data indicate that fatty acyl-CoAs may energetically favor the dimeric form of CtBP2 as their target.

**Dimerization is a key step for CtBP2 activation.** To further reinforce our hypothesis, we attempted to design constitutively active mutants of CtBP2 based on our structural modeling. Since CtBP2 forms dimers and multimers upon activation[24,39], we screened 79 amino acid residues in the CtBP2 protein facing the dimerization interface and calculated the predicted changes of protein stability and dimerization affinity for all of the possible mutations (1660 mutations in total) based on the difference of free energy (Fig. 8a and Supplementary Data 2). First of all, we generated 11 CtBP2 mutants with high scores for their stability and affinity in this in silico screen and examined the CtBP2/FoxO1 interaction, where the presence of ALA201HIS mutation most strikingly accelerated the CtBP2/FoxO1 interaction (Supplementary Fig. 9a). Indeed, increased dimer formation of this mutant was demonstrated by co-immunoprecipitation of differentially tagged constructs (Fig. 8b). The ALA201HIS mutant CtBP2 was relatively resistant against oleoyl-CoA-mediated inactivation although the mutation did not confer complete resistance (Fig. 8c). We further exogenously expressed this mutant CtBP2 along with wild-type CtBP2 in the liver of diet-induced obese mice (Supplementary Fig. 9b). The ALA201HIS mutant CtBP2 more potently ameliorated diabetes and hepatic steatosis than wild-type CtBP2 without inducing any changes in body weights (Fig. 8d, e and Supplementary Fig. 9c). As expected, the presence of this mutation increased the CtBP2/FoxO1 complex formation in vivo (Supplementary Fig. 9d) and potentiated the repressive activities of CtBP2 on the gluconeogenic and lipogenic programs (Fig. 8f, g) while the expression of non-specific gene (*Alb*) was not affected (Supplementary Fig. 9e). Here again, we further analyzed plasma lipid profile, and plasma total cholesterol was decreased by CtBP2 overexpression while plasma

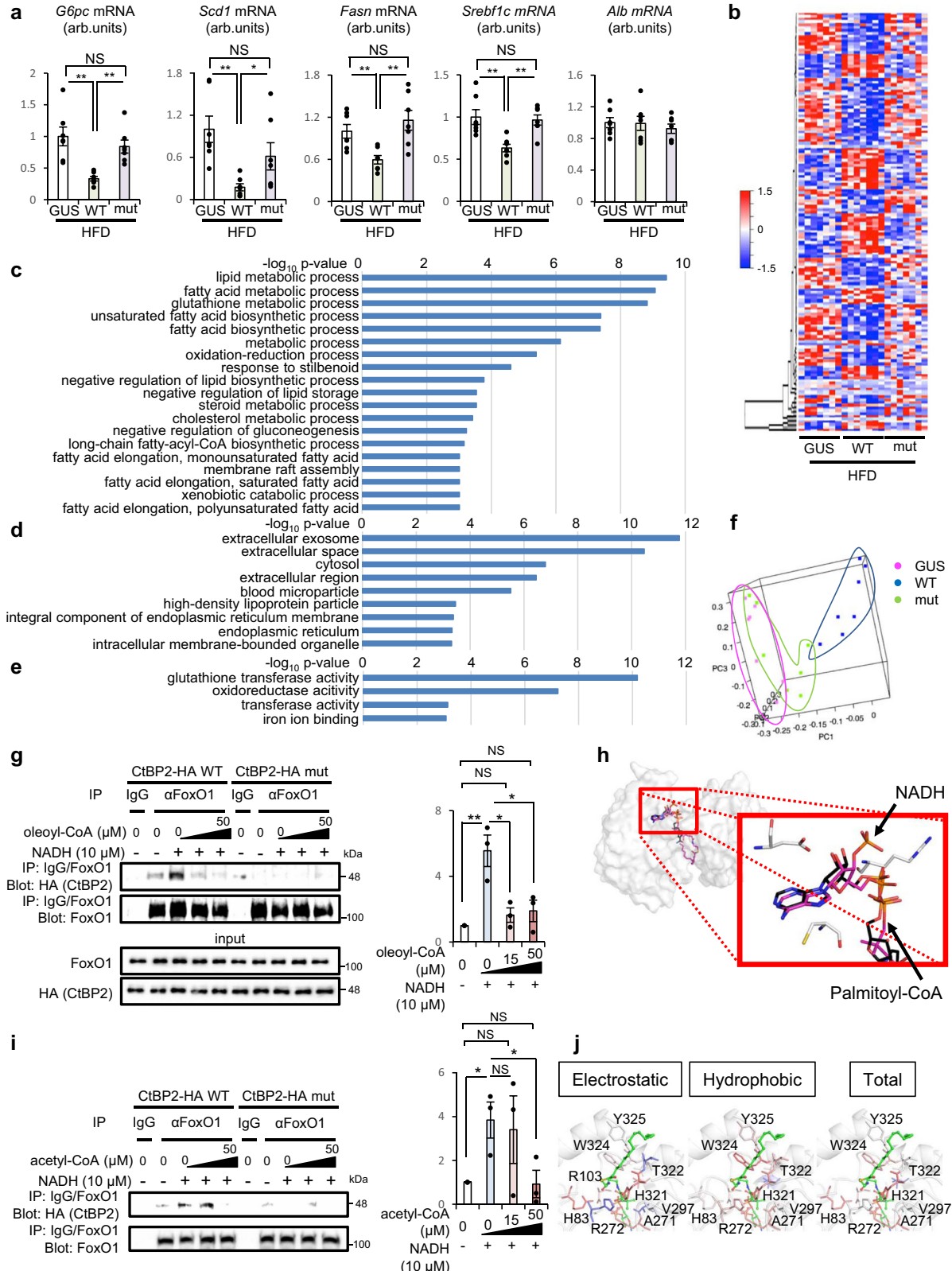

triglyceride was unchanged (Supplementary Fig. 9f). These metabolic benefits achieved with CtBP2 activation prompted us to measure energy expenditure of these mice that was tended to be increased by WT CtBP2 overexpression (Supplementary Fig. 9g).

Overall, these data support our model where CtBP2 integrates metabolic flux to control glucose and lipid homeostasis in liver (Fig. 8h). The metabolite imbalance between NADH/NAD$^+$ and

fatty acyl-CoA predisposes to obesity-associated metabolic disturbances through inactivation of CtBP2.

## Discussion
In this study, we identified a metabolite-driven system regulated by a corepressor CtBP2 that integrates metabolic fluxes to

**Fig. 7 The structure–function relationships in CtBP2 reveals critical coupling of metabolite-sensing with transcriptional regulation. a** Either wild-type CtBP2 (WT), CtBP2 mutated in its Rossmann fold (mut) or a control gene (GUS) was expressed in the liver of diet-induced obese (HFD) mice by adenoviral transduction and the expression of metabolic genes was analyzed ($n = 7$, biologically independent animals, 6-h fasted, p values for GUS vs WT/ WT vs mut are follows: $8.6 \times 10^{-4}/5.4 \times 10^{-4}$ (G6pc), $1.1 \times 10^{-3}/0.048$ (Scd1), $4.0 \times 10^{-3}/3.3 \times 10^{-3}$ (Fasn), and $2.9 \times 10^{-3}/8.2 \times 10^{-4}$ (Srebf1c), respectively). **b** Heatmap showing gene expression levels examined by RNA-seq. **c–e** Gene ontology analysis of RNA-seq data. Biological process (**c**), cellular component (**d**), molecular function (**e**). **f** Three-dimensional principal component analysis. Pink: GUS, blue: wild-type CtBP2, green: mutant CtBP2. **g** The CtBP2/FoxO1 complex formation was analyzed using purified recombinant proteins in vitro. Increasing concentrations of oleoyl-CoA (0, 15, 50 μM) with indicated concentrations of NADH (0 or 10 μM) were included. The p values are as follows: $9.0 \times 10^{-3}$ for control vs NADH+/oleoyl-CoA 0 μM, 0.021 for NADH+/oleoyl-CoA 0 μM vs NADH+/oleoyl-CoA 15 μM, 0.035 for NADH+/oleoyl-CoA 0 μM vs NADH+/oleoyl-CoA 50 μM. **h** Computer-assisted structural analysis of CtBP2 with NADH and palmitoyl-CoA. For comparison, images of NADH and palmitoyl-CoA were overlaid. **i** The effect of acetyl-CoA on CtBP2/FoxO1 interaction using recombinant proteins in the same setting as in **g**. The p values are as follows: 0.026 for control vs NADH+/acetyl-CoA 0 μM, 0.81 for NADH+/acetyl-CoA 0 μM vs NADH+/acetyl-CoA 15 μM, 0.048 for NADH+/acetyl-CoA 0 μM vs NADH+/acetyl-CoA 50 μM. Quantification of the recombinant protein interactions represented in Fig. 7g, i is shown to the right of each blot. Experiments were repeated three times for each. **j** Strucural modeling of CtBP2 complexed with palmitoyl-CoA by the first-principles calculations FMO. Part of the acyl-chain moiety is depicted in green. Data are expressed as the mean ± SEM. *, **, and NS denote $p < 0.05$, $p < 0.01$, and non-statistical significance evaluated by unpaired two-tailed Student's t-test, respectively. Source Data are provided as a Source Data file.

orchestrate glucose and lipid homeostasis in liver. Our findings redefined the role of CtBP2 as a sensor for multiple classes of metabolic intermediates rather than a sensor specifically designated for redox state. Furthermore, identification of FoxO1 as an obligate partner of CtBP2 opened an avenue for unraveling the metabolic response program orchestrated by CtBP2. Comprehensive resolution of CtBP2-mediated transcriptional programs by ChIP-seq and RNA-seq analyses convincingly demonstrated that the primary role of CtBP2 is regulation of metabolic pathways with profound dependence on its Rossmann fold. Importantly, inactivation of CtBP2 is the critical defect in the pathogenesis of obesity leading to metabolic inflexibility while activation of CtBP2 can be a therapeutic approach for the obesity-induced metabolic diseases. These findings led to a discovery that NAD(H) and fatty acyl-CoA, two metabolites without any direct connections reported, are inseparably connected through CtBP2 and the balance between these two metabolites can be a gauge reflecting cellular metabolic homeostasis.

The structure–function relationships revealed in this study clearly support our model where metabolite sensing in the Rossmann fold is tightly coupled with allosteric alterations of the CtBP2 activity that is regulated at the level of dimerization. Interestingly, fatty acyl-CoAs have two different sites of action to inhibit the dimerization: CoA moiety in the Rossmann fold pocket and long-chain fatty acyl moiety at the dimerization interface (Figs. 7h, j and 8h and Supplementary Movies 1 and 2). It is of note that CtBP2 preferably monitors NADH/NAD$^+$ and fatty acyl-CoAs among numerous adenosine derivatives since redox imbalance[18,19,47], excess fatty acid influx[7,16,48] resulting in accumulation of fatty acyl-CoAs[14,15] are hallmarks of a wide range of diseases.

While the repressor activity of CtBP2 seems to be regulated primarily at allosteric level upon binding to metabolic intermediates, the amount of CtBP2 also seems to be important. Our DSF assay also raised an important possibility that CtBP2 could be degraded even around our body temperature upon binding to fatty acyl-CoAs, which is of interest in terms of proteostasis (Fig. 2e). In this study, CtBP2 protein expression tended to be decreased in some obese models (Fig. 3e, i, Supplementary Fig. 4f and Supplementary Fig. 5c), which may be reflection of the fatty acyl-CoA-mediated CtBP2 protein degradation.

We exogenously expressed CtBP2 without changing hepatic metabolic intermediates in the liver of obese mice resulting in restoration of CtBP2/FoxO1 interaction. Although the affinity of CtBP2 for NAD$^+$ is much lower than that for NADH, NAD$^+$ still has capability to activate CtBP2 (ref. [20]) and the relative amount of NAD$^+$ is more than 100-fold higher than NADH in cells[49].

Therefore, exogenously expressed CtBP2 could be inefficiently activated by utilizing intracellular NAD$^+$ pool or other adenosine derivatives with low affinities[50].

We estimated the binding affinity of CtBP2 for NADH and oleoyl-CoA using MST. The $K_d$ of CtBP2 for oleoyl-CoA estimated in this study seems to be within the range of the physiological concentrations (~150 μM)[51] although there might be some technical limitations in both estimation of $K_d$ using MST and that of actual subcellular fatty acyl-CoA concentrations. Estimated $K_d$ of CtBP2 for NADH using MST in this study is higher than that reported previously with other assays (~100–1000 nM)[20,24,25]. This discrepancy may result from the differences of detection systems, protein concentrations applied (500 nM in our study is significantly lower than 2–20 μM in other studies), possible posttranslational modification of proteins (produced in insect cells in our study and prokaryotic expression system in others) or length of the recombinant protein (full-length CtBP2 in our study compared to CtBP2 lacking part of both termini in other studies). The temperature-gradient required for MST may have affected the stability of NADH. Since $K_d$ reflects both $K_{on}$ and $K_{off}$, higher $K_d$ does not necessarily indicate low binding affinity. If it is due to an increase in $K_{off}$, it would be more suitable as a metabolic sensor that quickly responds to metabolic alterations. The precise determination of $K_d$ may warrant further investigation. Regardless of this issue, the changes of thermophoretic properties induced by oleoyl-CoA in the MST assay was robust enough compared to the authentic ligand NADH, which supports our hypothesis together with the substantial thermal shifts observed in the DSF assay. It is of note that the DSF assay clearly demonstrated not only the direct binding of CtBP2 with both NADH and fatty acyl-CoAs but also the opposing effects on the conformation of CtBP2 induced by the binding of these two metabolites.

Transcriptional regulation is tightly coupled with the remodeling of chromatin architecture. Histones undergo a wide variety of acylation while the most preferred is acetylation or modifications with other short chain acyl derivatives[52], and the acetylation of histones decompacts chromatin to activate transcriptional programs[53]. In this sense, it is an intriguing observation that the acetylation donor, acetyl-CoA can derepress the gene repression mediated by a non-histone protein CtBP2, shedding light on the additional role of acetyl-CoA to activate transcriptional program. Moreover, while long-chain fatty acyl-CoAs have been under-explored in the transcriptional regulation, our study demonstrated that CtBP2 is the key molecule linking this acyl derivatives to transcriptional regulation. CtBP2 has been known to bind to its target transcription factors where CtBP2 recruits histone-

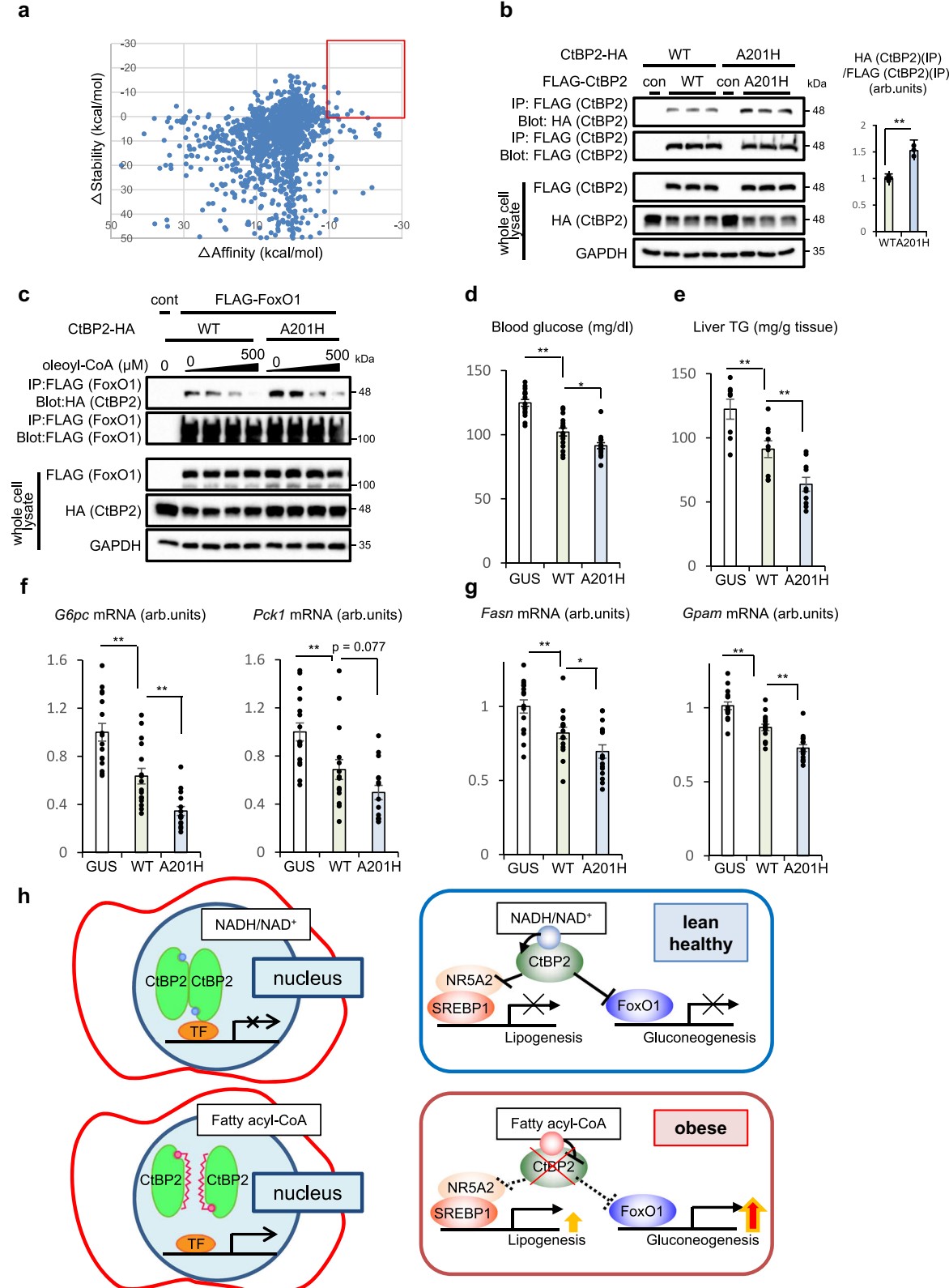

modifying enzymes such as HDACs (histone deacetylases), LSD1/ KDM1A (lysine-specific demethylase 1, lysine demethylase 1A), and EHMT1/2 (euchromatic histone methyltransferase 1/2) to alter the chromatin architecture[45]. Thus, long-chain fatty acyl-CoAs would remodel chromatin by modulating CtBP2 activity and altering recruitment of epigenomic modifiers at specific loci. Together with the fact that fatty acyl-CoAs are increased in obese

liver (Fig. 3m, n and ref. [43]), CtBP2 could be a key molecule linking the spill over of lipids in obesity to dysregulated metabolic programs. Indeed, exogenous expression of CtBP2 resulted in the deposition of repressive histone marks at the *G6pc* gene promoter where FoxO1 resides (Supplementary Fig. 7f), suggesting alteration of the chromatin architecture would be the key mechanism of CtBP2-mediated repression of FoxO1 and SREBP1. The

**Fig. 8 Dimerization is a key step for CtBP2 activation. a** Screening of mutations to stabilize the CtBP2 dimerization based on the computer-assisted structural analysis. y-axis: △stability, x-axis: △affinity. Red rectangle shows promising mutations to potentially promote CtBP2 dimerization. **b** Examination of dimer formation by co-immunoprecipitation of differentially tagged CtBP2 (interaction between FLAG-CtBP2 and CtBP2-HA, $n = 3$ biologically independent cells for each group, $p = 1.3 \times 10^{-3}$). **c** Cell lysates were obtained from HEK293 cells expressing either wild-type (WT) CtBP2 or ALA201HIS mutant (A201H) CtBP2 with FLAG-FoxO1, and the CtBP2/FoxO1 complex was detected in the presence of the increasing concentrations of oleoyl-CoA (0, 50, 150, 500 μM). **d–g** Adenovirus-mediated expression of either GUS, wild-type CtBP2 (WT), or ALA201HIS mutant CtBP2 (A201H) in diet-induced obese mice (3 days after transduction, overnight fasted). **d, e** Blood glucose levels (**d** $n = 16$, biologically independent animals, $p = 5.9 \times 10^{-6}$ and 0.012 for GUS vs WT and WT vs A201H, respectively) and liver triglyceride content (**e** $n = 8$, biologically independent animals, $p = 7.5 \times 10^{-3}$ and $4.7 \times 10^{-3}$ for GUS vs WT and WT vs A201H, respectively). **f, g** Expression levels of gluconeogenic (G6pc, Pck1, **f** $p = 8.2 \times 10^{-4}$ and $7.2 \times 10^{-4}$ for GUS vs WT and WT vs A201H (G6pc), $p = 8.8 \times 10^{-3}$ and 0.077 for GUS vs WT and WT vs A201H (Pck1), respectively) and lipogenic (Fasn, Gpam, **g** $p = 5.1 \times 10^{-3}$ and 0.047 for GUS vs WT and WT vs A201H (Fasn), $p = 1.6 \times 10^{-4}$ and $1.5 \times 10^{-4}$ for GUS vs WT and WT vs A201H (Gpam), respectively) genes ($n = 16$, biologically independent animals). **h** Schematic representation of a proposed model. Data are expressed as the mean ± SEM. * and ** denote $p < 0.05$ and $p < 0.01$ evaluated by unpaired two-tailed Student's t-test, respectively. Source Data are provided as a Source Data file.

components of the transcriptional complexes may vary in different contexts such as metabolic milieu, target transcription factors, and genomic loci that await further investigations.

The CtBP2/FoxO1 complex clearly responded to metabolic cues in physiological conditions: glucagon, glucocorticoid, insulin, and glucose in this study (Fig. 3a–d and Supplementary Fig. 4a). Interestingly and consistent with our observation, glucagon was reported to stimulate hepatic gluconeogenesis through an increase in fatty acyl-CoAs and acetyl-CoA generated from intrahepatic lipid mobilization[54]. Glucocorticoid may also increase cellular fatty acyl-CoAs through inhibition of mitochondrial activity[55] while insulin potently suppresses adipose lipolysis to decrease hepatic fatty acid influx resulting in a decrease of fatty acyl-CoAs. Therefore, these observations could be explained by the regulation of acyl-CoA concentrations. Glucose has been repeatedly reported to paradoxically increase expression levels of FoxO1-target genes with a potential mechanistic link to the oxidative stress response[55,56], which is also consistent with the regulation of CtBP2/FoxO1 complex observed in our study (Fig. 3d). Since increased glucose oxidation also inhibits fatty acid oxidation[57], increased fatty acyl-CoAs may underlie this observation as well. The net effects of these metabolic cues may explain our observation that the activity of CtBP2 is relatively maintained in response to the fasting feeding cycle (Fig. 3a). These observations seem to reflect the fact that cellular homeostasis is maintained with redundancy and compensatory mechanisms in physiological conditions represented here by fasting-feeding cycle while the activity of CtBP2 cannot be maintained in pathological conditions.

In the pathological conditions induced by obesity, CtBP2 dimer formation is disrupted upon binding to fatty acyl-CoAs resulting in dissociation from transcription factors. However, CtBP2 was reported to interact with some of its partners upon monomerization in a specific set of cases[58], suggesting interactions between CtBP2 and some transcription factors may be rather promoted in obesity. Thus, genetic deletion of CtBP2 may not faithfully mimic obesity-induced allosteric alterations of CtBP2. To address this issue, we may need to accumulate evidence from a wide range of different approaches including small molecule-mediated activation and inhibition of CtBP2.

Insulin stimulates hepatic de novo lipogenesis and suppresses gluconeogenesis in physiological conditions while both pathways are increased in obesity-associated pathological conditions[6], where it is an urgent clinical need to identify target molecule(s) to properly suppress both pathways as well as other pathology-associated pathways. In this sense, it is an intriguing observation that CtBP2 suppresses both pathways as well as metabolic inflammation[59,60] (Fig. 3j, Supplementary Fig. 1c and Supplementary Data 1) which is also supported by a recent report[61].

Together with the structure–function relationships, it is an attracting and plausible possibility to identify CtBP2-activating small molecule(s) targeting the Rossmann fold, which is currently under extensive investigation. This idea is further supported by a genome wide association study showing the association of a human CtBP2 gene variant with body mass index, implicating the relevance of CtBP2 in human obesity[62].

To explain the lipogenic pathway, we identified the CtBP2/NR5A2(LRH-1)/SREBP1 complex. The existence of an intermediary molecule, NR5A2(LRH-1), would confer multiple redundancies and complexities to the CtBP2-mediated repressive effect on SREBP1. The expression pattern of NR5A2(LRH-1), limited in specific tissues (liver, intestine, exocrine pancreas, and ovary[63]), would confer tissue-specific differences to the CtBP2/SREBP1 interaction, and NR5A2(LRH-1)/SREBP1 interaction would also be regulated irrespective of CtBP2 activity.

While we demonstrated the direct interaction between CtBP2 and FoxO1, FoxO1 activity has been reported to be regulated at the level of subcellular trafficking in response to hormonal stimuli[64]. Although supraphysiological concentrations of insulin may completely eliminate FoxO1 protein from the nucleus, some may remain on the DNA elements under non-extreme conditions as shown in our study (Fig. 4c and Supplementary Fig. 5b, c). It has been also shown that insulin inhibits the FoxO1 activity without nuclear exclusion[65,66]. Our data suggest that the regulation of FoxO1 activity may be more redundant than the predominantly accepted view. In addition, there also seems to be FoxO1-independent pathway(s) in the repression of hepatic gluconeogenesis by CtBP2 (Fig. 1e), that is consistent with our motif analysis of ChIP-seq peaks containing several gluconeogenic transcription factors (Supplementary Fig. 1d). Although our data suggest that CtBP2 functions mainly on the DNA element in the nuclei, we do not exclude other possibilities. Careful and sophisticated experimental approaches in the future may reveal other regulatory modes of CtBP2. While we focused CtBP2 between two CtBP isoforms in this study, CtBP1 may also have roles in metabolism. Despite these issues need to be resolved in the future, it is a ground-breaking discovery that CtBP2 confers capabilities to respond to intracellular metabolic alterations to metabolic transcription factors. Our findings support a central role of CtBP2 in global hepatic metabolic control and an exciting therapeutic potential of targeting this mechanism in metabolic diseases.

## Methods

**Animals.** All experimental procedures involving animals were conducted according to the guidelines of the Animal Care Committee, University of Tsukuba and the Institutional Animal Care and Use Committee, Harvard T.H. Chan School of Public Health, and all the experimental procedures were approved by these ethics committees. All mice used were males and maintained on a 12 h dark/light cycle

under standard conditions (23.5 ± 2.5 °C, 52.5 ± 12.5% humidity). Leptin-deficient *ob/ob* mice (10 weeks of age) and high-fat diet-induced obese mice (12–16 weeks on high-fat diet) were purchased from the Jackson Laboratories (Bar Harbor, ME). Mice were fed a MCD diet (Oriental Yeast, Tokyo) for a week from 7 weeks of age.

Recombinant adenoviruses encoding either β-glucuronidase (AdGUS), CtBP2 (AdCtBP2) were delivered into mice at a titer of $1.5 \times 10^{11}$ particles/mouse. In the study to examine ALA201HIS mutation, adenoviruses were delivered at a titer of $1.2 \times 10^{11}$ particles/mouse. Glucose tolerance tests and pyruvate tolerance tests were performed by intraperitoneal glucose (1 g/kg for mice on normal chow diet and 0.75 g/kg for diet-induced obese mice) and pyruvate (2 g/kg for mice on normal chow diet and 1 g/kg for diet-induced obese mice) administration following an overnight fast, and insulin tolerance tests were performed by intraperitoneal administration of insulin (0.25 U/kg for mice on normal chow diet and 0.5 U/kg for diet-induced obese mice) following 6-h food withdrawal.

The alteration of the activity of CtBP2 in response to metabolic cues were evaluated as follows: (1) Administration of either glucagon or insulin. We intraperitoneally administered either vehicle, glucagon (200 µg/kg) or insulin (0.4 U/kg) to wild-type mice on a regular diet following 3-h food withdrawal. After 30 min of the administration, the animals were euthanized and liver tissues were collected. (2) Glucocorticoid administration study. Wild-type mice were adrenalectomized and the successful adrenalectomy was confirmed by evaluating the serum corticosterone levels[67]. We intraperitoneally administered either vehicle or dexamethasone (1 mg/kg) to the adrenalectomized mice and liver tissues were collected after 6 h of the administration[67]. (3) Wild-type mice were given intraperitoneal injection of glucose (2 g/kg) following an overnight fast and liver tissues were collected after 30 min of the administration.

Serum ALT activity, total cholesterol, and triglyceride were analyzed using commercial kits (Sigma and Wako). Plasma insulin and glucagon levels were determined by ELISA kits (Levis and R & D systems, respectively). Liver triglyceride was extracted by the chloroform–methanol method and determined by a colorimetric assay (Sigma).

Energy expenditure was measured using the ARCO-2000 Mass Spectrometer System (Arcosystem, Inc). Oil Red O staining was performed as described previously[68].

**Generation of CtBP2-floxed mice**. We attempted to apply CRISPR/Cas9 system to generate mice with floxed allele for CtBP2 gene, and two sgRNAs were designed to introduce loxP sites into intron 4 and intron 6 of CtBP2 gene (the sgRNA sequences are follows: intron 4: actgacatccgctggcccgg, intron 6: ccatggaga-tagttgagcag). Since we could not obtain floxed allele by a single injection, we performed the second round of injection of sgRNA targeting intron 6 into zygotes carrying the loxP site in intron 4 generated by the first round of injection. We further crossed the CtBP2 flox mice with Alb-Cre transgenic mice to obtain liver-specific CtBP2-deficient mice.

**Primary hepatocyte isolation, analysis of transcription factors, and adenovirus transduction**. Primary mouse hepatocytes were isolated as described previously[69]. For adenoviral transduction, shRNA oligonucleotides were cloned into pENTR/U6 vector followed by the recombination with pAd/BLOCK-it-DEST vector (Thermo). The targeted sequences of shRNAs were designed as follows. Control: 5′-GTCTCCACGCGCAGTACATTT-3′, FoxO1: 5′-GCATGTTTATTG AGCGCTTGG-3′, NR5A2(LRH1): 5′-ACACAGAAGTCGCGTTCAAC-3′, CtBP2: 5′-GGGAAGACTAGGACGTGATTA-3′, and 5′-GCCACATTCTCAATCTGTA TC-3′. Forkhead response element (FHRE) luciferase (Addgene, Cambridge, MA) was cloned into pAd/PL-DEST vector (Thermo). Adenovirus encoding renilla luciferase was purchased from Vector Biolabs (Philadelphia, PA). Adenoviruses were amplified in HEK293A (R70507, Thermo) cells and purified by CsCl gradient centrifugation. FoxO transcriptional activity was measured in primary hepatocytes following transduction with adenoviruses expressing FHRE luciferase and renilla luciferase for 24 h. Cells were lysed and analyzed using Dual-Glo luciferase reporter system (Promega). Firefly luciferase signal was normalized to renilla luciferase.

**Quantitative real-time reverse transcriptase PCR**. Total RNA was isolated using Trizol Reagent (Thermo) and cDNA was synthesized with iScript Reverse Transcription Supermix (Bio-Rad). Quantitative real-time PCR analysis was performed using SYBR Green in ViiA 7 Real-Time PCR systems (Applied Biosystems) as previously described[70]. Data were normalized to acidic ribosomal phosphoprotein P0 (*Rplp0*, 36B4) expression. The primer sequences are shown in Supplementary Table 1.

**Nuclear fatty acyl-CoA content and liver lactate/pyruvate content**. Liver nuclei were isolated as described previously without detergent[69]. The lipids in the isolated nuclei were extracted basically by the Bligh-Dyer method using water acidified with acetic acid. Fatty acyl-CoA partitioned into methanol/water phase was collected. Fatty acyl-CoAs were oxidized by fatty acyl-CoA oxidase (Wako, Japan), yielding 2,3-trans-enoyl-CoAs and hydrogen peroxide. The hydrogen peroxide was quantified by a fluorescent detection system (Enzo Life Sciences, NY).

Liver tissues were deproteinated with 0.25 M metaphosphoric acid and the extracted lactate and pyruvate were measured using the fluorometric assay for L-lactate and pyruvate (Cayman Chemical).

**Western blot analysis and co-immunoprecipitation experiments**. Proteins were extracted from cells or liver samples with buffer A (50 mM Tris-HCl pH 7.4, 150 mM NaCl, 1% Nonidet P-40, 1 mM EDTA, 10 mM NaF, 2 mM $Na_3VO_4$) with complete protease inhibitors (Sigma-Aldrich) and subjected to SDS-polyacrylamide gel electrophoresis. Membranes were incubated with anti-FoxO1 (Cell Signaling, C29H4, 1:1000), anti-CtBP (Santa Cruz, E-12, 1:1000), anti-CtBP1 (BD, 612042, 1:1000), anti-CtBP2 (BD, 612044, 1:1000), anti-FLAG (Clontech, 635691, 1:2000), anti-GAPDH (Santa Cruz, FL-335, 1:2000), anti-SREBP1 (LSBio, LS-C179707, 1:1000), anti-ZFPM1 (Santa Cruz, A-6, 1:1000), anti-NR5A2(LRH1) (Novus Biologicals, NBP1-32489, 1:1000) anti-pPKA substrate (Cell Signaling, 100G7E, 1:1000), anti-LMNA (Cell Signaling, 2032, 1:1000), anti-Akt (Cell Signaling, C67E7, 1:1000), anti-pAkt (Ser 473, Cell Signaling, 9271, 1:1000), and anti-GUS (Thermo, MA1-35024, 1:1000). The membranes were incubated with the secondary antibody conjugated with horseradish peroxidase (Santa Cruz, sc-2004 (anti-rabbit IgG), sc-2005 (anti-mouse IgG)) and were visualized using the enhanced chemiluminescence system (Roche Diagnostics). To detect endogenous binding of FoxO1 and CtBPs, the anti-FoxO1 antibody (Santa Cruz, C-9), anti-CtBP antibody (Santa Cruz, E-12), anti-CtBP2 (Santa Cruz E-16, Active Motif 61261 or BD 612044), control mouse IgG (generated in house), or control goat IgG (Santa Cruz) were cross-linked to Protein G dynabeads (Thermo) with 50 mM dimethyl pimelimidate (Sigma-Aldrich). Primary hepatocytes or liver samples were lysed with buffer A and the protein complex was immunoprecipitated in buffer A with reduced concentration of NP40 (0.5%) (50 mM Tris-HCl pH 7.4, 150 mM NaCl, 0.5% Nonidet P-40, 1 mM EDTA, 10 mM NaF, 2 mM $Na_3VO_4$) for 4 h at 4 °C. The beads were washed with the buffer A with 0.5% NP40 four times, eluted with SDS loading buffer, and analyzed by western blot analysis. The FLAG-tag co-immunoprecipitation study was performed as follows. The plasmids encoding FLAG wild-type FoxO1-DsRed and CtBP1 are generous gifts from Domenico Accili (Columbia University, NY) and Pere Puigserver (Harvard Medical School, MA), respectively. CtBP2 cDNA was amplified by PCR and cloned into pcDNA3.1(+) (Thermo). The PSDL (Pro-Ser-Asp-Leu) motifs in FoxO1 and NR5A2(LRH1) were either mutated to PSAS (Pro-Ser-Ala-Ser) or deleted by gene tailor site-directed mutagenesis system (Thermo). The cytoplasmic FoxO1 mutant (△DB) was generated by deleting residues 205–217 of wild-type FoxO1 (ref. [44]).

HEK293 (CRL-1573, ATCC) cells were transiently transfected with control plasmid, FLAG wild-type FoxO1, or mutated FLAG FoxO1 along with either CtBP1 or CtBP2 using lipofectamine LTX (Thermo). Cells were lysed with buffer A with 1% NP40 and immunoprecipitated with FLAG M2 magnetic beads (Sigma) in buffer A with 0.5% NP40 for 4 h at 4 °C. The beads were washed four times with buffer A with 0.5% NP40 and eluted with 0.5 mg/ml of 3υ FLAG peptide (Sigma). To evaluate the effects of oleoyl-CoA (Sigma) and NADH (Sigma), the cell lysates were immunoprecipitated with FLAG M2 magnetic beads with increasing concentrations of oleoyl-CoA or NADH for 8 h at 4 °C. Thereafter, the FoxO1/CtBP2 complex was eluted and analyzed in the same way.

Transcriptional complex formation was also examined using purified recombinant proteins. We generated HEK293 cell lines stably expressing either FLAG-FoxO1, wild-type CtBP2-HA or mutant CtBP2 (G189A/G192A)-HA lacking the Rossmann fold[39] and recombinant proteins were affinity-tag purified. Those recombinant proteins were incubated with different concentrations of metabolites for 6–8 h at room temperature to examine the complex formation.

**Human liver specimens**. Human liver specimens from patients who underwent autopsy were purchased from ProteoGenex (Culver City, CA), with written informed consent obtained from all patients. Approval for this study was granted by the Medical Ethics Committee, University of Tsukuba.

**Subcellular localization study**. Hepa1-6 hepatoma cells were transfected with plasmids expressing FoxO1 tagged with DsRed (a gift from Domenico Accili) and/or CtBP2 tagged with GFP (OriGene) in the absence or presence of 100 nM insulin for 30 min. HEK293 cells were transfected with plasmids expressing either wild-type FoxO1-DsRed or △DB mutant FoxO1-DsRed along with CtBP2-GFP. Cells were fixed with 4% paraformaldehyde for 20 min and counterstained with DAPI.

**Chromatin immunoprecipitation**. Liver tissues were minced and fixed in 1% formaldehyde for 15 min at room temperature. Crosslinking was quenched by adding glycine to a final concentration of 125 mM. Thereafter, ChIP assay was carried out using Magna ChIP HiSens Chromatin Immunoprecipitation system (EMD Millipore) with minor modifications: chromatin shearing was achieved by micrococcal nuclease (Cell Signaling). Chromatin was immunoprecipitated either with control IgG (Cell Signaling), anti-CtBP2 (Active Motif, 61261), anti-FoxO1 (Abcam, ab39670), anti-acetylated H3K4 (Abcam, ab176799), anti-acetylated H3K9 (Cell Signaling, 9649), anti-acetylated H3K27(Active Motif, AM39133), and anti-H3K4me2 (Active Motif, AM39679). Immunoprecipitated DNA and input DNA were quantified by real-time PCR with primers specific for *G6pc*[71] or *Pck1* gene promoter along with negative control (primer sequence are as follows: *Pck1*

forward: 5′-tcggtcaacaggggaaatcc-3′, *Pck1* reverse: 5′-gaccctgcctacctttcttcctt-3′, *G6pc* forward: 5′-tggcttcaaggaccaggaag-3′, *G6pc* reverse: 5′-gatgcaaacatgttcagggtga-3′, or *G6pc* forward: 5′-atcaggctgtttttgtgtgcc-3′, *G6pc* reverse 5′-ggtgcatcatcag-taggttgatg-3′, negative control forward: 5′-cctctgtgcagctagaatggagt-3′, negative control reverse: 5′-ctgaagaattcatgagatgaggagag-3′). For sequential ChIP (re-ChIP), primary hepatocytes were transduced with FLAG-CtBP2 and FoxO1 adenoviruses. The chromatin was first enriched by FLAG magnetic beads (Sigma) and eluted with 3xFLAG peptide (Sigma). Afterwards, the second ChIP experiment was performed with either control goat IgG (Santa Cruz) or anti-FoxO1 antibody (Abcam, ab39670). The immunoprecipitated DNA was quantified using quantitative real-time PCR with the primers specific to *G6pc* gene promoter.

**ChIP-seq analysis and downstream data mining**. Liver samples were collected from mice fed normal chow diet following 6-h food withdrawal ($n = 4$). After cross-linking, 30 μg of chromatin DNA was immunoprecipitated with anti-CtBP2 antibody (Active Motif). A DNA fragment library for sequencing was constructed and the 75-nt sequence reads generated by Illumina sequencing (NextSeq 500) were mapped to the mouse genome using the BWA algorithm using the service of Active Motif. Total number of final reads was 38,476,546, and peaks were called using MACS algorithms. Genomic distribution of CtBP2-binding sites were analyzed using the *cis*-regulatory element annotation system (CEAS)[72]. Consensus motifs of CtBP2-binding sites were extracted using MEME (Multiple EM for Motif Elicitation)-ChIP program. To find candidate transcription factors bound to CtBP2, we further compared these motifs against databases of known motifs using TOMTOM (a motif comparison tool from MEME). Enrichment of gene ontology processes associated with the peaks was analyzed by GREAT (Genomic Regions Enrichment of Annotations Tool)[73]. Source: GEO Accession No. GSE127660.

**RNA-seq analysis and downstream data mining**. For RNA sequencing analysis, sequencing libraries were constructed from 500 ng total RNA using TruSeq RNA Sample Prep Kit v2 (Illumina) following the manufacturer's instruction. Libraries were validated with a Bioanalyzer (Agilent Technologies) to determine size distribution and concentration. Sequencing was performed by Tsukuba i-Laboratory LLP (Tsukuba, Japan) using NextSeq500 (Illumina) with paired-end 36-base read option. Sequencing reads were mapped to mm10 mouse reference genome and quantified using CLC Genomics Workbench version 9.5.1 (Qiagen). Fragments Per Kilobase of transcript per Million mapped reads (FPKM) values were estimated for each gene and filtered. A heatmap with hierarchical clustering was produced using the gplots and heatmap.2 packages in R. Gene ontology analysis and KEGG pathway analysis were performed using DAVID (The Database for Annotation, visualization and Integrated Discovery) software. Three-dimensional principal component analysis was conducted using the prcomp in R and visualized using rgl package. Source: GEO Accession No. GSE128145.

**Structural modeling of the CtBP2/palmitoyl-CoA complex**. The complex structure of human CtBP2 with palmitoyl-CoA was constructed using MOE program[74]. The crystal structure of the human CtBP2/NAD+/MTOB complex (PDB code: 4LCJ) in a dimer configuration was used as an initial model and protonated using Protonate 3D utilizing the AMBER10:EHT force-field with solvation energy accounted via the Born model. After removing NAD+ and MTOB from the human CtBP2/NAD+/MTOB complex, palmitoyl-CoA and acetyl-CoA were docked to the NAD+-binding site of human CtBP2 and the resultant complex structure was subjected to the energy minimization.

To evaluate molecular interaction between CtBP2 and acyl-CoA we performed the first-principles (quantum mechanical) calculations for the CtBP2/Palmitoyl-CoA complex based on the correlated fragment molecular orbital (FMO) method[75] at the resolution of the identity approximation of the second-order Møller-Plesset perturbation (RI-MP2) level[76] with the correlation-consistent polarized valence double zeta (cc-pVDZ) basis set[77] by using the PAICS program package (http://www.paics.net). The correlated FMO calculations could reliably evaluate not only electrostatic (ionic) but also weak CH-π (dispersion) interaction energies between the protein and the ligand/substrate[78]. In this study, molecular interaction energy between amino acid residue in CtBP2 and CoA molecule was analyzed by calculating inter-fragment interaction energies (IFIEs) under the FMO scheme. Electrostatic, dispersion, and the total energy of molecular interaction indicate the Hartree–Fock energy, ΔMP2, and the sum of these two interaction energies, respectively.

Dominant values of the IFIEs between amino acid residues in CtBP2 and Palmitoyl-CoA molecule were colored gradually by the 3D-VIEP plugin program[79]. The residues which formed stable and repulsive interactions were colored by red and blue, respectively. The magnitude of the IFIEs is correlated with color density. All of figures to represent CtBP2 structures were prepared by PyMOL[80].

**Molecular dynamics of the CtBP2/palmitoyl-CoA monomeric form and dimeric form**. CtBP2/palmitoyl-CoA dimeric form was constructed by Structure Alignment in Maestro (Schrödinger, LLC). The CtBP2/palmitoyl-CoA monomeric model was aligned on the A and B chains of the CtBP2 X-ray structure (PDB code: 4LCJ). Energy minimization calculation was performed on the two aligned

structures similar to the residue scanning method and minimized structures were used as the CtBP2/palmitoyl-CoA dimeric form. MD simulations was conducted using Desmond (Schrödinger, LLC). Monomeric form and dimeric form were placed in an orthorhombic box with a buffer distance of 10 Å to create a hydration model, and the SPC water model[81] was used for the hydration model. NaCl (0.15 M) served as the counter ion to neutralize the system. The cutoff radii for van der Waals and the time step, initial temperature, and pressure of the system were set to 9 Å, 2.0 fs, 300 K, and 1.01325 bar, respectively. The sampling interval during the simulation was set to 100 ps. Finally, we performed MD simulations using the NPT ensemble for 1 μs. All trajectories from MD simulation were aligned to the initial structure with protein Cα and ligand RMSF values were calculated based on ligand heavy atoms using Simulation Interactions Diagram in Maestro.

**Split luciferase complementation assay**. The CtBP2/FoxO1 interaction was detected basically using NanoLuciferase Binary Technology (BiT) system (Promega). NanoLuciferase employed in this assay is an engineered luciferase with advantageous properties such as smaller size (19 kDa) and higher luminescence. In the system, NanoLuciferase is devided into an 18 kDa large BiT and an 11 amino acid small BiT. These subunits were fused to either C-termni or N-termini of CtBP2 or FoxO1. Both CtBP2 and FoxO1 fused to either subunit were cloned into a single expression vector using the BiBiT Technology (Promega). All eight possible combinations (the subunit position (N-terminal or C-terminal) and the combination of CtBP2/FoxO1 and those subunits) were examined in transiently transfected HEK293 cells upon exposure to lactate. All of these combinations showed increased luciferase activities in response to lactate, and the cells with small BiT fused to C-terminal of CtBP2 and large Bit fused to C-terminal of FoxO1 showed highest luciferase activity. Therefore, clones stably expressing this combination were selected and used in this study.

**Microscale thermophoresis**. Human CtBP2 cDNA with strep-tag and HA-tag tandemly fused to the C-terminus was cloned into pFastBac HT vector (Thermo). The recombinant CtBP2 protein was produced using the baculovirus expression system (Thermo). To improve the purity of the recombinant protein, sequential purification using the nickel-NTA column and the Strep-Tactin column (IBA) was performed and the buffer was subsequently dialyzed into PBS. NADH and oleoyl-CoA MST ligand binding experiments were performed by 2bind GmbH (Regensburg, Germany). Fluorescently labeled CtBP2 protein was used at a constant concentration of 500 nM in assay buffer (20 mM Tris-HCl pH 8.0, 150 mM NaCl, 2 mM DTT, 0.05% Pluronic F-127, 0.085× PBS), the ligands were titrated from 5.4 to 165 nM (NADH) and 1 to 30.5 nM (oleoyl-CoA), respectively, in a serial dilution format. Measurements (two replicates with two runs each) were carried out at 60% MST power and 17% LED power in premium capillaries on a Monolith NT.115 device at 25 °C. Data were analyzed using MO.Affinity Analysis software. Data from corresponding experimental repeats were combined and fitted to a $K_D$ model derived from the law of mass action. CtBP2 protein was labeled using the Monolith Red-Tris-NTA dye (NanoTemper Technologies, Munich, Germany) with a twofold protein:dye excess (2 μM protein, 1 μM dye) in 20 mM Tris-HCl pH 8.0, 150 mM NaCl, 2 mM DTT, 0.05% Pluronic F-127, 0.17× PBS.

**Differential scanning fluorimetry**. DSF was carried out on a CFX qPCR system connected to CFX manager software (Bio-Rad). Either vehicle, NADH (20 μM final concentration), or oleoyl-CoA (20 μM final concentration) was added to the mixture of recombinant CtBP2 (0.2 mg/ml final concentration) and SYPRO Orange (Thermo) in a PBS-based buffer containing 137 mM NaCl, 2.7 mM KCl, 8.1 mM Na₂PO₄, and 1.5 mM KH₂PO₄ (pH 7.4). The PCR plate was heated from 20 to 90 °C with a step size of 1 °C min⁻¹, and fluorescence intensity was measured every 0.2 °C. The raw data were processed with CFX manager software, and the melting temperatures were determined by derivative analysis.

**Residue scanning**. To analyze differences in protein stability and dimer affinity by amino acid residue mutations, the X-ray structures of CtBP2 dimer (PDB ID:2OME) was downloaded from the Protein Data Bank (PDB). Assignment of bond orders and hydrogenation were performed using Maestro (Schrödinger, LLC). The suitable ionization states of NADH were generated by Epik[82] at pH $7.0 ± 2.0$. Hydrogen bond optimization was performed using PROPKA[83], and energy minimization calculations was conducted with Maestro using the OPLS3 force field[84]. Residue scanning wizard of BioLuminate (Schrödinger, LLC) was used to study the structural effects of mutations on the dimerization interface in optimized CtBP2 dimer structure.

**Statistical analysis and reproducibility**. Statistical analyses were carried out with a two-tailed unpaired Student's *t*-test. We repeated experiments to confirm reproducibility of our data at least two or three times for each. In addition, we attempted to verify our hypothesis and idea from different approaches of experiments (e.g. using recombinant proteins as well as cell lysates to observe the responses of CtBP2 to metabolites). Those preliminary data can be discussed upon reasonable requests.

## Data availability

RNA-seq and ChIP-seq datasets are available in the NCBI GEO database (GSE128145 (GSM3665562-3665582) and GSE127660 (GSM3636314-3636315), respectively. The reference genome for these analyses were GRCm38/mm10 and NCBI37/mm9, respectively. Unique materials (plasmids, viruses, and CtBP2 flox mice) are available upon reasonable requests from the corresponding author. All other data generated or analyzed during this study are included in this published article (and its supplementary information files). Source data are provided with this paper.

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

## Acknowledgements

M.S. conceived and initiated this project through works in the Department of Genetics and Complex Diseases, Harvard T.H. Chan School of Public Health, and the Sabri Ülker Center for Metabolic Research which were supported by Dr. Gökhan S. Hotamışlıgil. Our greatest acknowledgement goes to Dr. Hotamışlıgil. We thank the members of the Hotamışlıgil laboratory and Shimano laboratory for their contributions and invaluable discussions. We thank Gürol Tuncman, Ana Paula Arruda, Gunes Parlakgul, Benjamin Garfinkel, Lisa M. Rickey, Martin P. McGrath, and Kathryn C. Claiborn (Harvard University) for their support. We thank Dr. Gerald I Shulman (Yale University) for advice and support in measuring fatty acyl-CoA content. We thank Dr. Domenico Accili (Columbia University) for providing the FLAG-FoxO1-DsRed expression plasmid. We thank Dr. Pere Puigserver (Harvard Medical School) for providing CtBP1 expression plasmid. We thank Yang Ma, Kenji Saito, Takaaki Matsuda, Nanami Sekine, Kae Kumagai, Miwako Shobo (University of Tsukuba), Yuta Yamamoto, and Yurina Miyashita (Rikkyo University) for their help to characterize CtBP2. We thank Dr. Ryuichi Kato and Dr. Tatsuya Ohshida for recombinant protein production.

## Author contributions

M.S. conceived the idea, designed, and performed most of the experiments. K.K., S. Nagatoishi, and K.T. performed the DSF assay and other biomolecular interaction assays. T.S. helped preparation of CtBP2-knockout mice and obtained some preliminary data. Y.T. performed nuclear extraction from liver tissues. R.Y. and T.H. performed the in silico analysis to design constitutive active CtBP2 mutants as well as the MD simulation using monomeric form and dimeric form of CtBP2. H.T. and S. Nakano performed the in silico structural analysis. T. Miyamoto performed some imaging studies to support our hypothesis. T. Matsuzaka helped manuscript preparation and provided some animals. H.S. organized the project and helped the manuscript preparation. M.S. drafted the manuscript and incorporated all suggestions from the coauthors.

## Funding

M.S. was supported by a fellowship from the Manpei Suzuki Diabetes Foundation, the Advanced Research and Development Programs for Medical Innovation (PRIME) from the Japan Agency for Medical Research and Development (AMED) under Grant Number JP18gm5910007, Japan Society for the Promotion of Science, Takeda Science Foundation, Suzuken Memorial Foundation, Ono Medical Research Foundation, MSD Life Science Foundation and Novartis Foundation for the Promotion of Science. This research was partially supported by Platform Project for Supporting Drug Discovery and Life Science Research (BINDS) from AMED under Grant Number JP19am0101094.

## Competing interests

The authors declare no competing interests.

## Additional information

**Peer review information** *Nature Communications* thanks Shaodong Guo and the other anonymous reviewer(s) for their contribution to the peer review this work. Peer reviewer reports are available.

