## [Peer Review File · Nature Communications]

Reviewers' Comments:

Reviewer #1:

Remarks to the Author:

In this study, Dr. Sekiya and colleagues reported that the transcriptional co-repressor C-terminal binding protein 2 (CtBP2) acts as a NADH/NAD⁺ metabolite-sensor for modulating hepatic metabolism in mice. They showed that CtBP2 functions to inhibit FoxO1 activity and this effect contributes to the inhibition of gluconeogenesis in the liver. Likewise, CtBP2 acts to inhibit SREBP1 activity and this action contributes to the inhibition of lipogenesis in the liver. They further showed that dietary obese mice had lower CtBP2 protein levels in the liver. Liver-conditional CtBP2 depletion promoted both hepatic gluconeogenesis and lipogenesis, causing steatosis in mice. Adenovirus-mediated production of CtBP2 or its constitutively active mutant ameliorated diabetes and steatosis in dietary obese mice. Data from limited human liver autopsies (n=3-4) indicated that FoxO1-CtBP2 interaction was lower in obese subjects. Based on these data, the authors suggest that CtBP2 serves as a metabolite sensor for integrating intracellular NADH/NAD⁺ levels to hepatic glucose and lipid homeostasis.

Overall there are a lot of studies performed in vitro and in vivo, with data obtained from multiple animal models. However, the data are not quite coherent. There are many pieces of data derived from single measurements, without quantification and statistics. In particular, mechanistic insights are lacking in supporting the authors' conclusion that CtBP2 inhibits both FoxO1 and SREBP1 – two hepatic pathways that are differentially regulated in opposite manners. There are a number of specific comments for the authors to improve the manuscript.

1. Figure 1. CtBP2 is shown to inhibit FoxO1 activity via its interaction with the C-terminal domain of FoxO1. CtBP2 inhibition of FoxO1 is enhanced by insulin and diminished by glucose administration, without providing mechanistic insights. This observation seems at odds with the physiological state, in which glucose administration is coupled with acute insulin secretion from the pancreas in mice.
2. Likewise, CtBP2 acts to inhibit SREBP1 activity, without providing mechanistic insights. This mechanism seems important but appears counterintuitive. SREBP1 and FoxO1 are acting in opposite pathways, meaning that FoxO1 protein expression and activity are induced in fasted states, whereas SREBP1c protein expression and activity become activated in fed states. Physiologically, it is difficult to conceptualize the idea that CtBP2 functions to inhibit both SREBP1 and FoxO1 pathways, regardless of fasting or refeeding.
3. This also begs the question as to how CtBP2 expression is regulated in the liver and why CtBP2 is uniquely required for inhibiting two opposing pathways (FoxO1 in fasted phase and SREBP1 in fed state) in the liver?
4. Figure 3 panel g. Data from limited human liver autopsies (n=3-4) indicated that FoxO1-CtBP2 interaction was lower in obese subjects. Close examination of these data in Panel g does not strongly support the authors' conclusion. Neither FoxO1 nor CtBP2 protein levels were changed in lean vs. obese liver in human subjects. Fasting or fed states were not known either.
5. Figure 3 Panel I. Data in this panel appeared from a single mouse without statistics in each group.
6. Figure 4c. This panel shows that FoxO1 occupancy in the promoters of G6pc and Pck1 was similar in lean and HFD mice, contradicting the published data that FoxO1 activity, along with augmented gluconeogenic gene expression, is higher in the liver in obese mice. This panel's data are also at odds with the authors' idea that FoxO1 activity is higher in obese liver, due to the lack of CtBP2 inhibition on FoxO1 activity in obese mice (Figure Fig. 3e-i and Extended Data Fig. 4f). Please double-check these experiments to resolve these apparent self-conflicting results and controversies with data in the literature.
7. Figure 4b. It was not understandable why the authors chose a heterozygous HEK293 cell line (instead of a liver-based cell line) for ChIP assays. All other studies were performed on the mouse liver or hepatocytes.
8. Figure 4g. Despite a significant increase in fasting blood sugar levels and postprandial blood glucose levels, CtBP2-KO mice and WT had a similar fasting plasma insulin levels. This observation deserves a discussion.
9. Figure 4J and Figure 4m. CtBP2-KO mice developed mild steatosis on regular chow, and this effect was exacerbated on MCD diet. These effects ensued without changes in lipogenesis in the

liver. What is the underlying mechanism? What are blood triglyceride levels in CtBP2-KO mice? A significant induction of steatosis in the absence of altered lipogenesis could be derived from decreased VLDL secretion or reduced fatty acid oxidation in the liver – a possibility that was not attempted by the authors in the manuscript. Indeed, MCD diet feeding is known to cause steatosis by curbing hepatic VLDL-triglyceride secretion.

10. Figure 4. The authors did not determine hepatic FoxO1 or SREBP1 mRNA or proteins in the liver of CtBP2-KO mice. It remained unknown and unanswered whether hepatic CtBP2 depletion caused a significant increase in FoxO1 and SREBP1 activity, giving rise to augmented gluconeogenesis and increased lipogenesis in the liver of CtBP2-KO mice.

11. Figure 5. The authors used adenoviruses to increase CtBP2 expression in the liver of HFD obese mice. This experiment was based on the idea that CtBP2 is inactivated in the liver of obese mice. However, data presented in Figure 2 panels e and f did not show any significant alterations in liver CtBP2 protein levels in both ob/ob mice and HFD obese mice. Please confirm these studies.

12. Figure 5c. Despite the lack of a clear rationale for replenishing CtBP2 in obese liver, the authors showed that adenovirus-mediated CtBP2 production in the liver improved glucose tolerance. However, a puzzling observation is that obese mice with CtBP2 overproduction had blunt responses to pyruvate injection, indicating that CtBP2-overexpressed liver was impaired to undergo gluconeogenesis. Liver TG content was lower and blood glucose levels were lower in Adv-CtBP2 vector-treated obese mice. Where did the energy go and were blood lipid levels higher in Adv-CtBP2 vector-treated obese mice? It is desirable that the authors provide some explanation or insightful discussion for these observations. Ideally, the authors need to show that Adv-CtBP2 vector-treated obese mice might have increased energy expenditure and/or with a concomitant reduction in body weight gain. But the authors did not provide a reasonable clue to these observations.

13. Figure 5e. Oil Red O staining is required to better show the changes in hepatic lipid content in the liver. Please show these data in addition to H&E staining.

14. Figure 5. Fasting blood sugars in both control vector and Adv-CtBP2 vector-treated obese mice are relatively lower within a normal range. This might derive from a short duration of HFD feeding. Please acknowledge this point in the manuscript.

15. Figure 5. Please show the protein level of Adv-CtBP2 vector-mediated CtBP2 production in the liver over the control group. This information will help gauge how many folds of CtBP2 proteins were needed to achieve the beneficial effect on glucose and lipid metabolism in Adv-CtBP2 vector-treated obese mice.

16. In general, there is a lack of clarity about the state of mice (fasting or fed), in which ex vivo studies were performed in liver tissues.

17. Figure 7b. Data were based on a single measurement and were relatively weak in supporting the authors' conclusion, without quantification.

18. Figure 7. Overexpression of a constitutively active CtBP2 mutant in the liver corrected diabetes and reversed steatosis in obese mice, without changes in body weight. This conclusion needs more experimental data to support. For example, where did the energy go in this model? CtBP2 is presumed to inhibit both gluconeogenesis and lipogenesis in this model. What about the blood fatty acids and triglyceride levels? Does this model, as opposed to its controls, have altered energy expenditure?

19. Please limit the use of vague words such as "robustly" inactivated or "profoundly" decreased, etc, in the manuscript.

20. Please convert all bar graphs to dot-based graphs with individual dots representing individual mice for in vivo experiments or individual replicates for in vitro experiments. Dot-based graphs better illustrate the intra-subject variations and reproducibility.

21. Please indicate the molecular weight in all western blots.

22. Minor: Figure legends are not clear for readers.

23. Minor: The concept of CtBP2 was not well introduced in the Introduction section.

Reviewer #2:

Remarks to the Author:

The authors in this manuscript showed that Ctbp2 acts as a metabolite sensor in hepatic glucose and lipid homeostasis by supporting evidence that Ctbp2 cooperated with FoxO1 regulates metabolic genes in obese liver. There is no question about the quality and skills in this work, but,

the concept that Ctbp is a metabolic sensor has been already accepted in this field.

Major Concerns

- 1) Page 7 line 6-7: they mentioned that Ctbp1 is excluded in hepatic glucose metabolism through preliminary screening stage. Since Ctbp1, just like Ctbp2, acts as a transcriptional repressor in many cell types, they should provide the more reliable evidence that Ctbp1 is excluded in this experiment.
- 2) In Supplementary Figure 2e, they showed that FoxO1 mutant (Δ PSDL) did not alter the association with Ctbp1. As far as I know, Ctbp isoforms interact with substrate protein in a same manner, more sophisticated experiments may be needed. As shown in Figure 4f, mouse liver contains sufficient amount of both Ctbp1 and Ctbp2. Thus, without Ctbp1, all the data in this manuscript is incomplete to explain the FoxO1-Ctbp axis in hepatic glucose metabolism.
- 3) In Figure 2a, addition of 50 mM of NADH in cell lysates causes to increase the interaction of FoxO1 with Ctbp2. However, cell lysates have already contained the endogenous NADH (and NAD⁺), they should check the endogenous amount of NADH and calculate the ratio between endogenous NADH and exogenous NADH. NADH addition in cell lysates is different from the buffering system just like Figure 2b.
- 4) In Figure 3, they mentioned that Ctbp2 is markedly inactivated in the liver of obesity by showing that the dissociation between Ctbp2 and FoxO1 (other transcription factors) in several diet conditions. But these appears to be indirect. I'd like to point out that Ctbp2 itself does not have repressive activity. To strengthen their idea, they should provide the evidence of the dissociation of Ctbp (and FoxO1) with Ctbp2-associated repressors (such as Lsd1, Nurd complex) by IP experiment and ChIP experiments as well as the change of epigenetic marks (such as histone acetylation and so on) in the target genes.
- 5) In this manuscript, they generated the liver-specific Ctbp2 deficient mice (Figure 4f-m). I think this animal is good for analyzing the function of Ctbp2 in liver under various diet conditions. It would strengthen their idea.

Reviewer #3:

Remarks to the Author:

An important aspect of this manuscript is the structural model presented for the binding of palmitoyl CoA and acetyl CoA to CtBP2. The description of the structural modeling is quite superficial; if this reflects the actual structural modeling performed, that would be a cause for concern. The authors state that "palmitoyl-CoA and acetyl-CoA were docked into the NAD⁺ binding site of human CtBP2" and energy minimized. One would expect that the ADP substructure would fit in similarly for NAD⁺ and the CoA molecules, which provides a nice starting point. However, looking at the structure, it is not at all clear why the remainder of the CoA structures would continue down into the nicotinamide pocket, given the very different chemical structures, rather than exiting the interior of the protein as could easily occur near the phosphate groups. Based on examining the structures, it does not appear that palmitoyl-CoA would disrupt the dimeric assembly if it exited through this passage, whereas wiggling through the nicotinamide pocket to exit through the substrate pocket might. As the authors are pushing the idea that binding would lead to disruption of the dimeric assembly, how the docking was done and what choices were made becomes crucial. Unfortunately, based on the superficial description given, it is impossible to know if their model is likely or not, given the alternatives. Even if the authors are correct on this hypothesis, the presentation of the resultant models (Fig 2c and Fig. 6h) is not very illuminating. The authors also do not show a comparison of the packing of the palmitoyl-CoA with the location of the dimeric interface, which would be helpful for understanding the authors hypothesis that binding would disrupt the interface. It is worth noting that the CtBP dimer interface is very extensive (with buried surface area of over 2500Å²) so it would take substantial interactions from the acyl chain to disrupt it.

A central basis for considering that CtBP acts as a redox sensor is the conclusion from the Goodman group in the early 2000s that the affinity CtBP for NADH is 100 times greater than that of NAD⁺. (This work was done on CtBP1, so there is no evidence that CtBP2 binds NADH significantly tighter than NAD⁺ as the authors claim in their manuscript.) As far as I am aware, no

lab has been able to repeat/verify the conclusions of the Goodman group. The reported K_d for NADH binding in that work is similar to that reported by other methods of around 100-500nM. However, the reported K_d for NAD⁺ binding by a more indirect competition measurement of 8-11uM is much higher than any values reported by other groups. In addition to direct measurements of affinity (see, for instance, Madison et al. 2013, JBC), there are measurements of the response of CtBP to NADH and NAD⁺ in the oligomerization (Bellesis et al. 2018), and ability to stimulate binding of partners (Kumar et al 2002, Balasubramanian et al. 2002). None of these studies found evidence for substantially different responses of CtBP to NADH vs. NAD⁺. The interpretation of this manuscript rests heavily on a much higher affinity for NADH compared with NAD⁺, which is not supported by the preponderance of the data available.

The dissociation constant that the authors present using microscale thermophoresis between CtBP2 and NADH is two orders of magnitude higher (weaker binding) than reported by other methods. Although this is very briefly addressed in the appendix, I do not find their suggestions for the basis of this discrepancy very convincing, which adds additional concerns to the accuracy of their analysis.

Reviewers' comments:

Reviewer #1 (Remarks to the Author):

In this study, Dr. Sekiya and colleagues reported that the transcriptional co-repressor C-terminal binding protein 2 (CtBP2) acts a NADH/NAD⁺ metabolite-sensor for modulating hepatic metabolism in mice. They showed that CtBP2 functions to inhibit FoxO1 activity and this effect contributes to the inhibition of gluconeogenesis in the liver. Likewise, CtBP2 acts to inhibits SREBP1 activity and this action contributes to the inhibition of lipogenesis in the liver. They further showed that dietary obese mice had lower CtBP2 protein levels in the liver. Liver-conditional CtBP2 depletion promoted both hepatic gluconeogenesis and lipogenesis, causing steatosis in mice. Adenovirus-mediated production of CtBP2 or its constitutively active mutant ameliorated diabetes and steatosis in dietary obese mice. Data from limited human liver autopsies (n=3-4) indicated that FoxO1-CtBP2 interaction was lower in obese subjects. Based on these data, the authors suggest that CtBP2 serves as a metabolite sensor for integrating intracellular NADH/NAD⁺ levels to hepatic glucose and lipid homeostasis.

Response: We appreciate the Reviewer 1's careful evaluation of our manuscript. The Reviewer 1 raised a series of important points and we were able to improve our manuscript by responding to them. Some of them were unexpected, and we could not achieve this advancement without the Reviewer 1's suggestions.

Among the comments above, we would like to make sure that CtBP2 protein levels are relatively maintained even in the obese animals but that the activity of CtBP2 to bind to metabolic transcription factors was dramatically reduced in obesity. The Reviewer 1 would be aware of this, but we would like to emphasize that the function of CtBP2 is regulated at the levels of allosteric conformational changes, not at the levels of protein expression.

Overall there are a lot of studies performed in vitro and in vivo, with data obtained from multiple animal models. However, the data are not quite coherent. There are many piece of data derived from single measurement, without quantification and statistics. In

particular, mechanistic insights are lacking in supporting the authors' conclusion that CtBP2 inhibits both FoxO1 and SREBP1 – two hepatic pathways that are differentially regulated in opposite manners. There are a number of specific comments for the authors to improve the manuscript.

Response: We are afraid that our description was not sufficient for the Reviewer1 and readers to understand our complicated model. We would like to explain two major points before going to the point-by-point responses.

1) We would respectfully encourage the Reviewer 1 to understand our concept on the role of CtBP2 in physiological conditions. The activity of CtBP2 is relatively maintained stable in physiological conditions (Fig.3a), therefore contributions of CtBP2 activity to fasting feeding cycle would be relatively small compared to pathological conditions. In addition, CtBP2-mediated SREBP1 suppression requires intermediary molecule(s), LRH1 in our study or other molecules to be identified. Therefore, even when CtBP2 is active, CtBP2 would not be able to suppress SREBP1 without proper LRH1/SREBP1 complex formation. Here we show some data not included in our manuscript (shown below). These sets of data would hopefully help the Reviewer 1 to understand what we mentioned above. However, too much data and emphasis on physiology would make our description too disorganized to lead readers to understand our major point that is potential benefits of CtBP2 activation in obesity, a pathological condition. Considering the structural organization of our paper and relative contribution of CtBP2 activity in physiology, we would like to discuss this aspect elsewhere, not in this paper.

The responses of CtBP2 in physiology including CtBP2/SREBP1 interaction.

- 1) Glucagon has been known to activate gluconeogenesis and inhibit lipogenesis. Consistent with this, CtBP2 dissociates from FoxO1 and binds to SREBP1 by glucagon.
- 2) Glucocorticoid has been known to activate both gluconeogenesis and lipogenesis. Consistent with this, CtBP2 dissociates from both FoxO1 and SREBP1 by glucocorticoid although the latter is modest.
- 3) Insulin has been known to inhibit gluconeogenesis and activate lipogenesis. CtBP2 dissociates from FoxO1 but CtBP2/SREBP1 complex is unchanged by insulin.
- 4) Most importantly, CtBP2/FoxO1 and CtBP2/SREBP1 complexes are relatively maintained in the fasting feeding cycle, suggesting the contribution of CtBP2 activities to physiological changes would be limited. The discrepancies between CtBP2/FoxO1 and CtBP2/SREBP1 may be explained by the status of the intermediary molecule(s) that bridge CtBP2/SREBP1 interaction.

2) The Reviewer1 might be understanding in a way that more FoxO1 would be recruited in the promoters of gluconeogenic genes upon liberation from CtBP2's repression. However, what we propose is that CtBP2 forms a complex with FoxO1 bound to the promoters, and just dissociation of CtBP2 from the FoxO1 leads to upregulation of those genes while the amount of FoxO1 in those promoters are left unchanged upon liberation from CtBP2. All molecular events occur on the promoters. The Reviewer1's view would be that CtBP2/FoxO1 complex exists in cytosol or nuclei without binding to promoters and that the liberated FoxO1 from the complex would be recruited and bind to promoters upon liberation from CtBP2. We modified our schematic description (Fig. 8h) to clarify that CtBP2 dissociation from FoxO1 is occurring in the promoters and added a graphical description below to help the Reviewer1 to understand our proposed model.

Reviewer 1

A graphic description of our proposed model.

FoxO1 bound to the promoter with CtBP2 is inactive.

FoxO1 bound to the promoter without CtBP2 is active.

While nuclear-cytosolic shuttling could regulate FoxO1 activity to some extent, some FoxO1 remains on the promoter even in a fed state (Fig.4c, Extended Data Fig. 5b,c). CtBP2 targets FoxO1 bound to the promoter without influencing the shuttling system. Net effects of the shuttling system and CtBP2-mediated suppression (and other mechanisms) determine the FoxO1 activity.

1. Figure 1. CtBP2 is shown to inhibit FoxO1 activity via its interaction with the C-terminal domain of FoxO1. CtBP2 inhibition of FoxO1 is enhanced by insulin and diminished by glucose administration, without providing mechanistic insights. This observation seems at odd with the physiological state, in which glucose administration is coupled with acute insulin secretion from the pancreas in mice.

Response: The Reviewer is asking the role of CtBP2 in physiology and this comment may be related to Figure 3. Glucose administration may be accompanied with insulin secretion, however, administration of either glucagon, insulin or glucose would emphasize the effect of each despite the overall response may reflect metabolic changes evoked secondarily. We showed the effect of fast/fed to have a fair assessment of physiological state without emphasizing specific metabolic cues where we did not observe robust alterations of the CtBP2 activity. Our main claim here is that CtBP2 activity is relatively maintained in a physiological state whereas it is markedly inactivated in obesity. We also described possible mechanisms behind this in our

supplementary discussion which may have been missed due to its presence in supplementary discussion. We included the discussion in the main text.

As the Reviewer 1 pointed out, what we expected was that glucose administration would increase the CtBP2/FoxO1 interaction. However, what we observed was opposite but consistent with the literature cited in our discussion that proposes FoxO1 activation in response to glucose to cope with oxidative stress (page 27, line 7-12). Since glucose utilization inhibits fatty acid oxidation, increased fatty acyl-CoA may explain this observation as we discussed. CtBP2 regulates part of the FoxO1 activity and we would like to emphasize that CtBP2/FoxO1 interaction does not necessarily correlate with total FoxO1 activity especially in physiological conditions.

2. Likewise, CtBP2 acts to inhibit SREBP1 activity, without providing mechanistic insights. This mechanism seems important but appears counterintuitive. SREBP1 and FoxO1 are acting in opposite pathways, meaning that FoxO1 protein expression and activity are induced in fasted states, whereas SREBP1c protein expression and activity become activated in fed states. Physiologically, it is difficult to conceptualize the idea that CtBP2 functions to inhibit both SREBP1 and FoxO1 pathways, regardless of fasting or refeeding.

Response: Here again the activity of CtBP2 is relatively maintained stable in both fasted and fed states, indicating relative contributions of CtBP2 to alterations of gluconeogenic and lipogenic gene expression in physiology would be limited. Inactivation of CtBP2 in obesity may in part explain the concomitant activation of gluconeogenesis and lipogenesis in obese liver. In addition, CtBP2 regulates SREBP1 indirectly. Thus, the intermediary molecule, LRH1 in our manuscript or unidentified molecule(s), may also play roles in lipogenic gene expression in this context. CtBP2 activation would directly repress gluconeogenesis irrespective of LRH1 status but CtBP2 would repress lipogenesis in a LRH1-dependent manner as we described in our discussion. To help the Reviewer 1 to understand this complexity, we provided data above mentioned. Interaction of CtBP2 with FoxO1 is not always in parallel with that with SREBP1 in physiology that may be explained by the presence of intermediary molecule(s) or other mechanisms. We can further investigate this aspect, but relative contributions of CtBP2 activities to physiological changes (i.e. fasting feeding cycle) are

limited, therefore, we would like to focus on pathological conditions in this report. Too much data to clarify the details in physiology would not be friendly for broad readership in this journal, we believe.

3. This also begs the question as to how CtBP2 expression is regulated in the liver and why CtBP2 is uniquely required for inhibiting two opposing pathways (FoxO1 in fasted phase and SREBP1 in fed state) in the liver?

Response: Regarding the expression of CtBP2, we showed it in Extended Data Fig. 5c. What we have observed is that protein expression of CtBP2 is relatively maintained compared to the drastic alterations of CtBP2 activity in obesity. We would emphasize that functional binding ability of CtBP2 is diminished in obesity and that protein expression of CtBP2 is relatively maintained.

Based on our observations, CtBP2 expression at mRNA and protein levels may respond to physiological and pathological conditions but the extent of alterations is relatively small and we cannot draw a consistent conclusion through multiple experiments. Therefore, we would like to focus on the CtBP2 activity in this study that is consistent throughout multiple experiments. We would like to avoid showing data with uncertainties and we believe we can propose our model without determining the relatively small fluctuation of CtBP2 expression since the CtBP2 activity predominantly determines the effects of CtBP2.

4. Figure 3 panel g. Data from limited human liver autopsies (n=3-4) indicated that FoxO1-CtBP2 interaction was lower in obese subjects. Close examination of these data in Panel g does not strongly support the authors' conclusion. Neither FoxO1 nor CtBP2 protein levels were changed in lean vs. obese liver in human subjects. Fasting or fed states were not known either.

Response: These are human autopsy samples and we cannot control fasted/fed state and quality of samples. Those samples were collected hours after donors died. We already tried to improve the quality of data, but we cannot expect high quality for these human samples compared to those from experimental animals. If this set of data can be an obstacle, we would withdraw the data, which would not influence rigidity of our

proposal. Again, we would like to emphasize that CtBP2 activity shown in the co-immunoprecipitation experiments are reduced and that we don't necessarily need to observe reduced protein expression of CtBP2.

5. Figure 3 Panel I. Data in this panel appeared from a single mouse without statistics in each group.

Response: We increased the number of samples with statistics (new Fig.3I).

6. Figure 4c. This panel shows that FoxO1 occupancy in the promoters of *G6pc* and *Pck1* was similar in lean and HFD mice, contradicting the published data that FoxO1 activity, along with augmented gluconeogenic gene expression, is higher in the liver in obese mice. This panel's data are also at odd with the authors' idea that FoxO1 activity is higher in obese liver, due to the lack of CtBP2 inhibition on FoxO1 activity in obese mice (Figure Fig. 3e-i and Extended Data Fig. 4f). Please double-check these experiments to resolve these apparent self-conflicting results and controversies with data in the literature.

Response: The Reviewer1 raised quite an important point. The question whether or how much FoxO1 occupancy in the *G6pc* promoter is altered in response to physiological and pathological conditions is not completely settled although the insulin-dependent nuclear cytosolic shuttling has been predominantly accepted. Although we don't need to argue against this shuttling view to explain our proposal, we can find several reports contradictory to this view.

1) Sci Rep. 2016 Oct 19;6:35531. doi: 10.1038/srep35531.

LncRNA SRA promotes hepatic steatosis through repressing the expression of adipose triglyceride lipase (ATGL)

Gang Chen¹, Dongsheng Yu², Xue Nian², Junyi Liu³, Ronald J Koenig⁴, Bin Xu⁴, Liang Sheng²

They quantify nuclear FoxO1 contents in diet-induced obesity and genetic obesity in both fasted and fed conditions in figure 3b,c to show no alteration of nuclear FoxO1 contents in these conditions.

2) Br J Nutr. 2012 Jul;108(2):218-228. doi: 10.1017/S0007114511005563. Epub 2011 Oct 20.

Regulation of glucose metabolism via hepatic forkhead transcription factor 1 (FoxO1) by *Morinda citrifolia* (noni) in high-fat diet-induced obese mice

Pratibha V Nerurkar¹, Adrienne Nishioka¹, Philip O Eck¹, Lisa M Johns^{#1}, Esther Volper^{#2}, Vivek R Nerurkar²

In this paper, they quantify nuclear FoxO1 contents in diet-induced obesity in Figure 3. They claim that high fat diet feeding increased nuclear FoxO1 contents but we cannot recognize the difference as far as we see the actual blots.

3) Front Physiol. 2018 Jan 23;9:15. doi: 10.3389/fphys.2018.00015. eCollection 2018.

Astragaloside IV Inhibits Adipose Lipolysis and Reduces Hepatic Glucose Production *via* Akt Dependent PDE3B Expression in HFD-Fed Mice

Qun Du¹, Shuihong Zhang², Aiyun Li³, Imran S Mohammad⁴, Baolin Liu², Yanwu Li¹

This paper supports the historically accepted shuttling view. They show a clear increase of FoxO1 in response to high fat diet feeding in Figure 6.

As far as we checked literature, no existing paper clearly quantified FoxO1 occupancy in *G6pc* promoter. We cannot find any ChIP-seq datasets showing FoxO1 occupancy in that promoter in obesity. We also cited two papers showing the shuttling-independent FoxO1 activation in our manuscript.

1) X. Zhang *et al.*, Phosphorylation of serine 256 suppresses transactivation by FKHR (FOXO1) by multiple mechanisms. Direct and indirect effects on nuclear/cytoplasmic shuttling and DNA binding. *The Journal of biological chemistry* **277**, 45276-45284

(2002).

2) W. C. Tsai, N. Bhattacharyya, L. Y. Han, J. A. Hanover, M. M. Rechler, Insulin inhibition of transcription stimulated by the forkhead protein Foxo1 is not solely due to nuclear exclusion. *Endocrinology* **144**, 5615-5622 (2003).

Although the shuttling system of FoxO1 has been accepted as an exclusive mechanism, we may need to be careful about the relative contribution of this system based on the literature and our unpublished data.

What we propose is that loss of CtBP2 binding liberates FoxO1 and promotes gluconeogenic gene transcription. In this system, we don't need to observe increased FoxO1 in those promoters. Loss of CtBP2 repression with the same amount of FoxO1 left in those promoters will increase the downstream gene expression. The Reviewer1 may be taking our model in a way that FoxO1 is recruited into promoters upon liberation from CtBP2. To clarify this issue, we modified schematic description of our model in new Fig. 8h and provided a graphic description in the response described above.

7. Figure 4b. It was not understandable why the authors chose a heterozygous HEK293 cell line (instead of a liver-based cell line) for ChIP assays. All other studies were performed on the mouse liver or hepatocytes.

Response: We repeated the same experiment using mouse primary hepatocytes (new Fig. 4b).

8. Figure 4g. Despite a significant increase in fasting blood sugar levels and postprandial blood glucose levels, CtBP2-KO mice and WT had a similar fasting plasma insulin levels. This observation deserves a discussion.

Response: The Reviewer1 is again raising an important point. Although we have not identified possible systemic effects on insulin biology induced by liver-specific CtBP2 deletion, we don't deny that possibility based on our preliminary data. We added a statement in our Results section (page 17, line 2-5).

9. Figure 4J and Figure 4m. CtBP2-KO mice developed mild steatosis on regular chow,

and this effect was exacerbated on MCD diet. These effects ensued without changes in lipogenesis in the liver. What is the underlying mechanism? What are blood triglyceride levels in CtBP2-KO mice? A significant induction of steatosis in the absence of altered lipogenesis could be derived from decreased VLDL secretion or reduced fatty acid oxidation in the liver – a possibility that was not attempted by the authors in the manuscript. Indeed, MCD diet feeding is known to cause steatosis by curbing hepatic VLDL-triglyceride secretion.

Response: This is quite an important suggestion but we again omitted a set of data to keep readers on the main path. The extent of alterations of plasma lipids induced by CtBP2 deletion in liver was modest, but the lipid fluxes would at least in part contribute to phenotype. We added data in Extended Data Fig. 6h, k. In addition, plasma cholesterol levels are negatively correlated with CtBP2 activity in liver, that would be informative for pharmacologists looking for new druggable targets for dyslipidemia considering the broad readership of the journal.

10. Figure 4. The authors did not determine hepatic FoxO1 or SREBP1 mRNA or proteins in the liver of CtBP2-KO mice. It remained unknown and unanswered whether hepatic CtBP2 depletion caused a significant increase in FoxO1 and SREBP1 activity, giving rise to augmented gluconeogenesis and increased lipogenesis in the liver of CtBP2-KO mice.

Response: We again appreciate the Reviewer 1's suggestion. We assumed the expression levels of these two transcription factors were unchanged, but in fact both FoxO1 and SREBP1 expression levels were reduced in CtBP2 knockout mice (new Fig. 5g, Extended Data Fig. 6i). This would be an additional compensatory change and would explain the modest phenotype observed in the knockout mice compared to that observed in the overexpression models.

11. Figure 5. The authors used adenoviruses to increase CtBP2 expression in the liver of HFD obese mice. This experiment was based on the idea that CtBP2 is inactivated in the liver of obese mice. However, data presented in Figure 2 panels e and f did not

show any significant alterations in liver CtBP2 protein levels in both ob/ob mice and HFD obese mice. Please confirm these studies.

Response: We may have had to explain more carefully although we have a statement in our discussion. What we are proposing is that obese mice have decent amount of CtBP2 protein but that is functionally inactivated. CtBP2 in obesity cannot bind to FoxO1 due to the functional deterioration. We firstly overexpressed wild-type CtBP2 to activate it although it does not faithfully mimic metabolite-dependent CtBP2 activation. Therefore, we added a study using a CtBP2 mutant which mimic functional activation in Figure 8. We need to identify small molecules to activate CtBP2 metabolite-sensing pocket to completely convince readers. This will be out of scope of this paper, but we indeed identified and have it in our hands with expected metabolic benefits. We hope the amended schematic description of our model in Fig.8h would facilitate proper understanding.

12. Figure 5c. Despite the lack of a clear rationale for replenishing CtBP2 in obese liver, the authors showed that adenovirus-mediated CtBP2 production in the liver improved glucose tolerance. However, a puzzling observation is that obese mice with CtBP2 overproduction had blunt responses to pyruvate injection, indicating that CtBP2-overexpressed liver was impaired to undergo gluconeogenesis. Liver TG content was lower and blood glucose levels were lower in Adv-CtBP2 vector-treated obese mice. Where did the energy go and were blood lipid levels higher in Adv-CtBP2 vector-treated obese mice? It is desirable that the authors provide some explanation or insightful discussion for these observations. Ideally, the authors need to show that Adv-CtBP2 vector-treated obese mice might have increased energy expenditure and/or with a concomitant reduction in body weight gain. But the authors did not provide a reasonable clue to these observations.

Response: We will discuss this in the response to Q18 that is same as this question.

13. Figure 5e. Oil Red O staining is required to better show the changes in hepatic lipid content in the liver. Please show these data in addition to H&E staining.

Response: We included Oil Red O staining in Extended Data Fig. 7g. Thanks to this comment, we were able to highlight a substantial difference induced by CtBP2 overexpression.

14. Figure 5. Fasting blood sugars in both control vector and Adv-CtBP2 vector-treated obese mice are relatively lower within a normal range. This might derive from a short duration of HFD feeding. Please acknowledge this point in the manuscript.

Response: We changed the description according to this suggestion (page 18, line 14-15).

15. Figure 5. Please show the protein level of Adv-CtBP2 vector-mediated CtBP2 production in the liver over the control group. This information will help gauge how many folds of CtBP2 proteins were needed to achieve the beneficial effect on glucose and lipid metabolism in Adv-CtBP2 vector-treated obese mice.

Response: The corresponding data is shown in Extended Data Fig.7a

16. In general, there is a lack of clarity about the state of mice (fasting or fed), in which ex vivo studies were performed in liver tissues.

Response: We already indicated those conditions in Materials and Methods as well as in some of the figure legends. In this version, we included all of the information in the figure legends.

17. Figure 7b. Data were based on a single measurement and were relatively weak in supporting the authors' conclusion, without quantification.

Response: We increased the number of samples with statistics (new Fig. 8b).

18. Figure 7. Overexpression of a constitutively active CtBP2 mutant in the liver corrected diabetes and reversed steatosis in obese mice, without changes in body weight. This conclusion needs more experimental data to support. For example, where

did the energy go in this model? CtBP2 is presumed to inhibit both gluconeogenesis and lipogenesis in this model. What about the blood fatty acids and triglyceride levels? Does this model, as opposed to its controls, had altered energy expenditure?

Response: We provided plasma lipid profiles (new Extended Data Fig. 9f) as well as energy expenditures (new Extended Data Fig. 9g) in these animals. But if we roughly calculate mass turnover, liver lipid reduction would be around $(50 \text{ mg/g tissue}) \times (\sim 1.5 \text{ g liver/mouse}) = \sim 75 \text{ mg}$. If we assume the amount of whole body plasma $\sim 1 \text{ ml}$, triglyceride in blood would be $(\sim 100 \text{ mg/dl}) \times (1 \text{ ml}) = 1 \text{ mg}$. Of course, we need to take the turnover of lipids into account, these are trivial compared to the body weights ($\sim 40\text{g}$). This is also the case for blood glucose (we can roughly convert glucose/lipid based on the formula 4 kcal/g for glucose and 9 kcal/g for lipid). Although we agree that CtBP2 activation may have a potential to reduce body weights, we may need longer period of time to observe significant reduction of body weights, which would not be evaluated by the transient adenovirus-mediated overexpression.

However, the Reviewer 1 is raising an important question again as he/she asked a potential role of CtBP2 in insulin biology. We have preliminary data supporting this idea in another model that will be discussed in future publications.

19. Please limit the use of vague words such as “robustly” inactivated or “profoundly” decreased, etc, in the manuscript.

Response: This kind of suggestion is quite useful and informative for non-native scientists like us. We amended our manuscript according to the Reviewer1’s suggestion

20. Please convert all bar groups to dot-based graphs with individual dots representing individual mice for in vivo experiments or individual replicates for in vitro experiments. Dot-based graphs better illustrate the intra-subject variations and reproducibility.

Response: We amended our graphs according to the Reviewer1’s suggestion.

21. Please indicate the molecular weight in all western blots.

Response: We amended our blots according to the Reviewer1's suggestion.

22. Minor: Figure legends are not clear for readers.

Response: We amended our figure legends according to the Reviewer1's suggestion.

23. Minor: The concept of CtBP2 was not well introduced in the Introduction section.

Response: We added some statements in our Introduction section. One is the existence of controversy regarding the redox sensing capability of CtBP2 based on the differential affinities for NADH and NAD⁺. The other is non-transcriptional function of CtBP1.

Reviewer #2 (Remarks to the Author):

The authors in this manuscript showed that Ctbp2 acts as a metabolite sensor in hepatic glucose and lipid homeostasis by supporting evidence that Ctbp2 cooperated with FoxO1 regulates metabolic genes in obese liver. There is no question about the quality and skills in this work, but, the concept that Ctbp is a metabolic sensor has been already accepted in this field.

Response: We sincerely appreciate the Reviewer 2's acknowledgement of our quality. Regarding the novelty of our findings, CtBP2 as an NADH sensor is widely accepted, but no existing paper demonstrated that it is a fatty acyl-CoA sensor. Importantly, inactivation of CtBP2 in obesity seems to be induced by the increased fatty acyl-CoA, not by NADH/NAD⁺ ratio, suggesting fatty acyl-CoA sensing is one of the major biological roles of CtBP2. While previous studies attempted to understand CtBP2 in the 'NADH vs NAD⁺' context, our model in this study is rather that CtBP2 is a sensor working in the 'NADH vs fatty acyl-CoA' context. Indeed, the Reviewer 3 pointed out that our core finding in this manuscript would be the fatty acyl-CoA sensing by CtBP2. This idea would provide new perspectives to understand our metabolic system and may

solve some controversy regarding NADH/NAD⁺ sensing function of CtBP2 that was asked by Reviewer 3. We amended our statements to make these things clearer. Moreover, our findings indicate that the balance between NADH/NAD⁺ and fatty acyl-CoA, two metabolites without any direct connections reported, can be a novel gauge for cellular metabolic homeostasis, that can be the beginning of a new paradigm. In addition, the link between CtBP2 and metabolic transcription factors is also novel, which redefine CtBP2 as a central player in a new metabolic system. The metabolic system regulated by CtBP2 is composed of a lot of novel interactions and idea. We refrained from repeating the word 'novel' in our manuscript, but please let us list our major novel findings here. We believe our manuscript provide a conceptual change supported by a lot of novel findings although it is always hard to assure 'novelty'.

- 1) CtBP2 interacts with FoxO1 and FoxO1-mediated regulation of gluconeogenesis is under the control of CtBP2.
- 2) CtBP2 adopts acyl-CoAs (long chain fatty acyl-CoA and acetyl-CoA).
- 3) Structural modeling of CtBP2 with acyl-CoA.
- 4) Negative regulation of CtBP2 by fatty acyl-CoA. Previous papers reported that NADH activates CtBP2 more efficiently than NAD⁺ but that both NADH and NAD⁺ activate CtBP2. Therefore, the identification of fatty acyl-CoA as a negative regulator of CtBP2 can offer a drastic conceptual change.
- 5) CtBP2 activity is dramatically altered in liver tissues in physiological and pathological conditions.
- 6) CtBP2 interacts with SREBP1 and SREBP1-mediated regulation of lipogenesis is under the control of CtBP2. In addition, we identified the intermediary molecule LRH1.
- 7) Generation of CtBP2 flox mice and the phenotype of liver specific CtBP2 knockout mice: diabetes and hepatic steatosis and liver damage in a specific condition. It is of note that concurrent activation of hepatic gluconeogenesis and lipogenesis in obesity has been a long-lasting mystery in this field.
- 8) Liver-specific overexpression of CtBP2 ameliorates diabetes and hepatic steatosis.
- 9) Identification of A201H mutation that facilitates CtBP2 dimer formation.
- 10) Based on these findings, we proposed that CtBP2 plays a critical role in liver glucose and lipid metabolism and could be targeted to develop future medicine.
- 11) The balance between NAD(H) and fatty acyl-CoA can be a novel indicator for metabolic homeostasis.

Major Concerns

1) Page7 line 6-7: they mentioned that Ctbp1 is excluded in hepatic glucose metabolism through preliminary screening stage. Since Ctbp1, just like Ctbp2, acts as a transcriptional repressor in many cell types, they should provide the more reliable evidence that Ctbp1 is excluded in this experiment.

Response: As far as we examined, CtBP1 seems to be involved in our metabolic system as well. However, it did not seem to be straightforward (shown below). For example, knockdown of CtBP1 and CtBP2 had synergistic effects on gluconeogenic gene expression *in vitro* whereas knockdown of CtBP1 and CtBP2 *in vivo* provided different expression profile. We believe that we need to understand the basis for this discrepancy and the metabolic roles of CtBP1 seems to be more complicated. If the roles of CtBP1 and CtBP2 were just synergistic, we may describe them in a single paper, but we think it would be too broad to have both CtBP isoforms in a single paper. We also added a statement regarding the possible role of CtBP1 in metabolism as a future avenue in our Discussion (page 29, line15-17). In addition, we clearly stated that CtBP2 isoform may have 'direct' involvement rather than CtBP1 isoform in our Results section (page 7, line 7) since our original statement may give an impression that CtBP1 does not have any metabolic effects.

Reviewer 2

in vitro

CtBP1 and/or CtBP2 were suppressed in primary hepatocytes and expression levels of *Pck1* (shown above) and *G6pc* (data not shown) were analyzed. Both CtBP1 and CtBP2 have synergistic effects on gluconeogenic genes *in vitro*.

in vivo

Knock-down of either CtBP1 or CtBP2 in the liver of normal mice. While CtBP2 suppression increased gluconeogenic and lipogenic genes, CtBP1 suppression did not affect expression of these genes.

2) In Supplementary Figure 2e, they showed that FoxO1 mutant (Δ PSDL) did not alter the association with Ctbp1. As far as I know, Ctbp isoforms interact with substrate protein in a same manner, more sophisticated experiments may be needed. As shown in Figure 4f, mouse liver contains sufficient amount of both Ctbp1 and Ctbp2. Thus, without Ctbp1, all the data in this manuscript is incomplete to explain the FoxO1-Ctbp axis in hepatic glucose metabolism.

Response: We think that CtBP1 also binds to FoxO1 through its PSDL motif directly and indirectly, and the contribution of indirect binding is strong enough to minimize the effect of deletion of the direct binding. Again, if we demonstrate the roles of CtBP1 as well as CtBP2 in this paper, the amount of data would be tremendous. We believe it would be friendly for readers if we focus on CtBP2 in this paper.

3) In Figure 2a, addition of 50 mM of NADH in cell lysates causes to increase the interaction of FoxO1 with Ctbp2. However, cell lysates have already contained the endogenous NADH (and NAD⁺), they should check the endogenous amount of NADH

and calculate the ratio between endogenous NADH and exogenous NADH. NADH addition in cell lysates is different from the buffering system just like Figure 2b.

Response: At first, we need to discriminate between bound and free NADH/NAD⁺ since CtBP2 would be under the influence of free NADH/NAD⁺. Since it is really hard to determine free NADH/NAD⁺ concentrations in cell lysates, it is often estimated by indirect measurements. In addition, we are using an exaggerated system where CtBP2 and FoxO1 are overexpressed with decent concentrations of NADH. Free NADH would be ~100 nM based on the cited paper (K. A. Anderson, A. S. Madsen, C. A. Olsen, M. D. Hirschey, Metabolic control by sirtuins and other enzymes that sense NAD(+), NADH, or their ratio. *Biochimica et biophysica acta* **1858**, 991-998 (2017)), endogenous NADH would not affect the final concentrations of NADH in this experiment.

We also showed a more simplified system using recombinant proteins and metabolites (Fig.7g,i), which would sufficiently convince readers.

4) In Figure 3, they mentioned that Ctbp2 is markedly inactivated in the liver of obesity by showing that the dissociation between Ctbp2 and FoxO1 (other transcription factors) in several diet conditions. But these appears to be indirect. I'd like to point out that Ctbp2 itself does not have repressive activity. To strengthen their idea, they should provide the evidence of the dissociation of Ctbp (and FoxO1) with Ctbp2-associated repressors (such as Lsd1, Nurd complex) by IP experiment and ChIP experiments as well as the change of epigenetic marks (such as histone acetylation and so on) in the target genes.

Response: We appreciate this quite valuable suggestion. Since general obesity influences a number of molecules and some of them would be CtBP2-independent, we evaluated histone codes modified by CtBP2 overexpression to have a fair assessment on the effects of CtBP2 regarding this issue (new Extended Data Fig.7f). Modifications of histone marks are indispensable when we evaluate the function of CtBPs. This is quite an important question.

5) In this manuscript, they generated the liver-specific Ctbp2 deficient mice (Figure 4f-m). I think this animal is good for analyzing the function of Ctbp2 in liver under

various diet conditions. It would strengthen their idea.

Response: We challenged those mice with an MCD diet in this manuscript. High fat diet and other obesogenic diet would inactivate CtBP2 and the difference between wild-type and CtBP2-KO would disappear. We also tried a streptozotocin-induced type 1 diabetes model. We did not include this data since it would not add new insights over existing data. If the reviewer requires additional model(s), we can add the data.

Reviewer #3 (Remarks to the Author):

An important aspect of this manuscript is the structural model presented for the binding of palmitoyl CoA and acetyl CoA to CtBP2. The description of the structural modeling is quite superficial; if this reflects the actual structural modeling performed, that would be a cause for concern. The authors state that “palmitoyl-CoA and acetyl-CoA were docked into the NAD⁺ binding site of human CtBP2” and energy minimized. One would expect that the ADP substructure would fit in similarly for NAD⁺ and the CoA molecules, which provides a nice starting point. However, looking at the structure, it is not at all clear why the remainder of the CoA structures would continue down into the nicotinamide pocket, given the very different chemical structures, rather than exiting the interior of the protein as could easily occur near the phosphate groups. Based on examining the structures, it does not appear that palmitoyl-CoA would disrupt the dimeric assembly if it exited through this passage, whereas wiggling through the nicotinamide pocket to exit through the substrate pocket might. As the authors are pushing the idea that binding would lead to disruption of the dimeric assembly, how the docking was done and what choices were made becomes crucial. Unfortunately, based on the superficial description given, it is impossible to know if their model is likely or not, given the alternatives. Even if the authors are correct on this hypothesis, the presentation of the resultant models (Fig 2c and Fig. 6h) is not very illuminating. The authors also do not show a comparison of the packing of the palmitoyl-CoA with the location of the dimeric interface, which would be

helpful for understanding the authors hypothesis that binding would disrupt the interface. It is worth noting that the CtBP dimer interface is very extensive (with buried surface area of over 2500Å²) so it would take substantial interactions from the acyl chain to disrupt it.

Response: We appreciate the Reviewer 3's comments since we would have readers with similar impression upon publication of our original manuscript. The acyl-chain moiety extends out of the pocket, pivots with fixation to the pocket through its CoA moiety and finally locates at the dimerization interface with hydrophobic and electrostatic interactions. We provided another image from a different angle to clarify the location of palmitoyl-CoA in CtBP2 dimeric assembly (Extended Data Fig. 8f). And of note, not only the acyl-chain moiety at the dimerization interface but also CoA moiety play a pivotal role by competing the pocket with NADH.

We further performed the first-principles calculation using fragment molecular orbital (FMO) method to quantify the interaction energies (Fig.7j) and Extended Data Fig. 8g) and this analysis was further validated by a mutation study (Extended Data Fig. 8h). If the Reviewer 3 requires more solid data such as X-ray crystallography or cryo-electron microscopy, that should be reported separately.

A central basis for considering that CtBP acts as a redox sensor is the conclusion from the Goodman group in the early 2000s that the affinity CtBP for NADH is 100 times greater than that of NAD⁺. (This work was done on CtBP1, so there is no evidence that CtBP2 binds NADH significantly tighter than NAD⁺ as the authors claim in their manuscript.) As far as I am aware, no lab has been able to repeat/verify the conclusions of the Goodman group. The reported K_d for NADH binding in that work is similar to that reported by other methods of around 100-500nM. However, the reported K_d for NAD⁺ binding by a more indirect competition measurement of 8-11uM is much higher than any values reported by other groups. In addition to direct measurements of affinity (see, for instance, Madison et al. 2013, JBC), there are measurements of the response of CtBP to NADH and NAD⁺ in the oligomerization (Bellesis et al. 2018), and ability to stimulate binding of partners (Kumar et al 2002, Balasubramanian et al. 2002). None of these studies found evidence for substantially different responses of CtBP to NADH vs. NAD⁺. The interpretation of this manuscript rests heavily on a much higher affinity for NADH

compared with NAD⁺, which is not supported by the preponderance of the data available.

Response: We thank the Reviewer 3 raising this controversy. We evaluated the effect of NADH/NAD⁺ ratio in Figure 2f and 2i indirectly using live cultured cells. In most of our studies in this manuscript, we evaluated the effects of NADH without any comparison to those of NAD⁺. And we also stated the binding of NADH seems to be relatively unstable compared to fatty acyl-CoA in Figure 2b. As mentioned above, one of our major points in this study is that CtBP2 is a fatty acyl-CoA sensor. In fact, we showed inactivation of CtBP2 in two different obese mouse models that cannot be explained by NADH/NAD⁺ ratio (Extended Data Fig.4g,h). Our proposal is that CtBP2 is serving as a sensor in the context of NADH/fatty-acyl-CoA ratio rather than NADH/NAD⁺ ratio, which would solve the controversy raised by the Reviewer 3. We attempted to clarify this by adding statements in our Introduction (page 5, line 18-20), Results (page 23, line12-14) and Discussion (page 24, line 14-18) sections. We completely agree with the existence of this controversy which needs to be solved. There are a lot of possibilities, for instance, purified CtBP protein alone may not be able to distinguish NADH and NAD⁺ and some cellular environments may be required. Indeed, we can observe redox sensitivities in cultured cells (Fig. 2f,2i) that cannot be observed in purified proteins in agreement with the literature provided by the Reviewer 3 (please refer to the data below). Although this

Reviewer 3

Examination of binding kinetics using SPR technology.

In this assay, the estimated K_d values for NADH and NAD were 5-50 nM range while that for oleoyl-CoA was 10 mM-10 M range (we repeated experiments and representative images are shown).

We obtained K_d values for NADH and NAD⁺ similar to those reported whereas that for oleoyl-CoA was quite high due to unidentified experimental limitations.

The estimated K_d values were not consistent with those obtained by MST and DSF, that may be explained by the immobilization specifically required to SPR technology.

There are a lot of systems to evaluate binding affinities and each system has some inevitable limitations.

question is of great interest for scientists in a broad field as well as for us, we can construct our proposed model in this study without solving this issue.

The dissociation constant that the authors present using microscale thermophoresis between CtBP2 and NADH is two orders of magnitude higher (weaker binding) than reported by other methods. Although this is very briefly addressed in the appendix, I do not find their suggestions for the basis of this discrepancy very convincing, which adds additional concerns to the accuracy of their analysis.

Response: No existing reports evaluated binding kinetics of CtBP2 using microscale thermophoresis. Since there might be technical pitfalls in this experimental system, we additionally examined this aspect by different systems. We attempted to estimate the K_d using the differential scanning fluorimetry (DSF) assay (Extended Data Fig. 3c in the revised manuscript), where we observed similar K_d values for both NADH and oleoyl-CoA. We also attempted to measure the kinetics using Biacore technology, a surface plasmon resonance (SPR) biosensor system, where K_d for NADH and oleoyl-CoA were 5-50 nM range and 10 mM-10 M range (please refer to the data shown

above), respectively. We observed a discrepancy between MST/DSF and SPR in terms of the estimated K_d values. The major difference would be that SPR requires immobilization of recombinant proteins and that MST and DSF provide more natural conditions for molecules to interact. Although we showed data obtained through MST and DSF because of the relatively natural conditions of the experiments and consistency, we don't deny potential difficulties in determination of binding kinetics. Although every experiment may have some possible limitations, we showed multiple layers of evidence at the levels of protein, cultured cells and *in vivo* models to support our conclusion. We moved our corresponding discussion into the main text as well. We as well as other scientists in this field have not performed any unbiased screening experiments to identify the most biologically relevant ligand(s) for CtBP2. Thus, no one knows if NADH is exclusively regulating CtBP2 activity. We also added a statement to imply potential contribution of non-NADH/NAD⁺ ligand(s) in our discussion (page 25, line 19). We believe we can only accumulate evidence through many experimental conditions that will eventually lead us to the truth. We hope our data could be part of that accumulated evidence.

Reviewers' Comments:

Reviewer #2:

Remarks to the Author:

There are no more questions to this manuscript.

I believe that they responded to reviewer's concerns as possible as they could.

Reviewer #3:

Remarks to the Author:

I appreciate the responses to my earlier concerns and accept their revisions regarding CtBP as a redox sensor. However, I am still concerned about the lack of detail concerning the molecular modeling to understand the palmitoyl-CoA binding to CtBP2. Although I appreciate the additional figure (extended Fig. 8f), which does do a better job of showing their model, I am not convinced that this model is accurate. In reading through the results and methods, I can't even tell if they were working with a monomer of CtBP2 or a dimer. (I expect a monomer, but it should be stated.) One interesting approach would be to run the calculations with both a monomer and a dimer, to see if there are alternate pathways for the acyl-chain moiety. In my opinion, this is a serious flaw, but recognize that there is a lot more in the paper.

Reviewer #4:

Remarks to the Author:

In this study, the authors investigated the role of CtBP2, a metabolite sensor, in hepatic glucose and lipid homeostasis. Under obese condition, CtBP2 is inactivated majorly through the increased fatty acyl-CoA, its ability to bind to Foxo1 and Srebp1 is diminished, and its suppressive effect on Foxo1 and SREBP1 is attenuated. CtBP2 gain-of-function studies showed that exogenously expressed CtBP2 improves the glucose tolerance and hepatic steatosis in obese mice. Overall, there are two major findings in this study. One is fatty acyl-CoA binds to CtBP2 and inhibits its activity; the other is CtBP2, as a transcriptional corepressor, interacts with Foxo1 directly and SREBP1 indirectly to suppress their transcriptional activity, thereby CtBP2 has a beneficial effect on glucose homeostasis and lipid profile. However, the description of the mechanisms by which CtBP2 inhibits Foxo1 and SREBP1 activity is not clear. I suggest that the authors add the molecular mechanism of CtBP2 inhibiting Foxo1 and SREBP1 into discussion part.

There are several questions as following:

1. Whether fatty acyl-CoA diminishes the interaction between CtBP2 and SREBP1? I did not find the results supporting this.
2. The author mentioned that CtBP2 protein level in the liver is barely changed. However, in Figure 3E and I it shows that CtBP2 protein level is largely downregulated in obese mice livers. Please double check this.
3. It is weird that the molecular weight of Flag-Foxo1 is more than 100 kDa, since the MW of FLAG is only 1 kDa. Please double check.

Comments on the authors' response to Reviewer 1

1. One of the major concerns from Reviewer 1 is that Foxo1 and SREBP1 pathways are regulated in opposite manners in liver. It is difficult to conceptualize the idea that CtBP2 functions to inhibit both Foxo1 and SBREBP1 pathways. The authors responds that the activity of CtBP2 is relatively maintained in both fasting and random-fed conditions. Although CtBP2 inhibits both Foxo1 and SREBP1 pathways, the mechanism is kind of different. CtBP2 binds to Foxo1 directly and inhibits its activity. CtBP2 binds to SREBP1 through LRH1 indirectly and inhibits its activity. In my opinion, it may be true that CtBP2 does not play a key role in regulation of Foxo1 and SREBP1 pathways under physiological state, whereas other mechanisms are involved. The reason might be the key metabolites regulating CtBP2 activity, such as NADPH and fatty acyl-CoA, are maintained at a reasonable range. However, under a pathological condition (obesity), owing to the significant increase of fatty acyl-CoA, the activity of CtBP2 is inhibited. Foxo1 and SREBP1 pathways are

activated, which impairs glucose and lipid homeostasis. To some extent, the role CtBP2 may explain the selective insulin resistance in diabetes.

2. Another important question the Reviewer 1 raised is that the Foxo1 occupancy in the promoters of G6pc and Pck1 is similar in lean and HFD mice. I am not sure it is true or not, since Foxo1 protein level should be upregulated in HFD mouse livers and accordingly the Foxo1 occupancy in its target genes' promoters should be also increased. However, these results do not contradict to the rationale the authors proposed. They propose that CtBP1 directly inhibits Foxo1 activity without affecting its occupancy in the promoters of G6pc and Pck1. If so, another interesting question is how CtBP1, as a transcription repressor, inactivates Foxo1.

In general, the authors answered the reviewer 1's questions. The most questions of review 1 is generated from the molecular mechanisms of CtBP1. In this manuscript, the authors did not clearly describe the molecule mechanisms of CtBP1. Thus, I suggest the authors add this part into the discuss

Reviewers' comments:

Reviewer #2 (Remarks to the Author):

There are no more questions to this manuscript.

I believe that they responded to reviewer's concerns as possible as they could.

Response: We appreciate Reviewer #2 who indeed selected the most important points in this manuscript.

Reviewer #3 (Remarks to the Author):

I appreciate the responses to my earlier concerns and accept their revisions regarding CtBP as a redox sensor. However, I am still concerned about the lack of detail concerning the molecular modeling to understand the palmitoyl-CoA binding to CtBP2. Although I appreciate the additional figure (extended Fig. 8f), which does do a better job of showing their model, I am not convinced that this model is accurate. In reading through the results and methods, I can't even tell if they were working with a monomer of CtBP2 or a dimer. (I expect a monomer, but it should be stated.) One interesting approach would be to run the calculations with both a monomer and a dimer, to see if there are alternate pathways for the acyl-chain moiety. In my opinion, this is a serious flaw, but recognize that there is a lot more in the paper.

Response: We appreciate comments from Reviewer #3 that kindly remind us of missing pieces in our structural modeling.

We started our structural modeling of CtBP2 in a dimer configuration that was deposited in the PDB (Protein Data Bank, PDB code 4LCJ) in the previous version of our manuscript. In this revision, we amended our description to clearly state that it is a dimer (page 43, line 15). In addition, we added another structural simulation where we observed interactions between palmitoyl-CoA and either monomeric CtBP2 or dimeric CtBP2 (Extended Data Fig. 8i, Supplementary Movie1 and 2). Interestingly, the CoA moiety was captured in the Rossmann fold pocket in both monomeric and dimeric CtBP2 while the acyl-chain was structurally

stabilized only in the presence of dimeric configurations. In other words, the acyl-chain moiety requires the dimeric interface to be fixed. The RMSF measurement that we used to quantify the fluctuation of palmitoyl-CoA residues indicates that palmitoyl-CoA may energetically favor dimer CtBP2 as its target. We believe that this experiment clearly demonstrated that the acyl-chain locates at the dimerization interface, and we really thank the Reviewer #3 for proposing this experiment.

Reviewer #4 (Remarks to the Author):

In this study, the authors investigated the role of CtBP2, a metabolite sensor, in hepatic glucose and lipid homeostasis. Under obese condition, CtBP2 is inactivated majorly through the increased fatty acyl-CoA, its ability to bind to Foxo1 and Srebp1 is diminished, and its suppressive effect on Foxo1 and SREBP1 is attenuated. CtBP2 gain-of-function studies showed that exogenously expressed CtBP2 improves the glucose tolerance and hepatic steatosis in obese mice. Overall, there are two major findings in this study. One is fatty acyl-CoA binds to CtBP2 and inhibits its activity; the other is CtBP2, as a transcriptional corepressor, interacts with Foxo1 directly and SREBP1 indirectly to suppress their transcriptional activity, thereby CtBP2 has a beneficial effect on glucose homeostasis and lipid profile. However, the description of the mechanisms by which CtBP2 inhibits Foxo1 and SREBP1 activity is not clear. I suggest that the authors add the molecular mechanism of CtBP2 inhibiting Foxo1 and SREBP1 into discussion part.

Response: We really appreciate Reviewer #4 for his/her proper understanding of our manuscript irrespective of our complicated proposed model. We moved our supplementary discussion to the main text and added a few sentences to clarify that CtBP2 binds to transcription factors such as FoxO1 and SREBP1 where CtBP2 remodels chromatin structures, that was also experimentally supported by the Extended Data Figure 7f. Thanks to this suggestion, we were able to describe a sequential event where alteration of the interaction between CtBP2 and transcription factors by metabolites leads to chromatin remodeling and transcriptional repression (for example, page 28 line 20-page 29 line 2).

There are several questions as following:

1. Whether fatty acyl-CoA diminishes the interaction between CtBP2 and SREBP1? I did not find the results supporting this.

Response: We appreciate this suggestion. We added Extended Data Fig. 4i that fills in the missing piece in our manuscript.

2. The author mentioned that CtBP2 protein level in the liver is barely changed. However, in Figure 3E and I it shows that CtBP2 protein level is largely downregulated in obese mice livers. Please double check this.

Response: We really appreciate the Reviewer #4's careful review of our manuscript. We completely agree with this comment. Although we have been recognizing this decreased CtBP2 protein expression in obese liver (Fig. 3e, 3i, Extended Data Fig. 4f, 5c), we avoided overstatement because of the imperfect reproducibility (Fig. 3f, 3h). We believe that CtBP2 protein expression is also decreased in obesity contributing to our proposed model to some extent, and we may be able to solve this issue. At this moment, we added a statement to faithfully describe the presented data in the Discussion section (page 26, line 12-14).

3. It is weird that the molecular weight of Flag-Foxo1 is more than 100 kDa, since the MW of FLAG is only 1 kDa. Please double check.

Response: Again, we appreciate the Reviewer #4's careful review. We used a plasmid encoding FLAG-FoxO1 fused to DsRed, a gift from Dr Domenico Accili. We clarified this in sections of Acknowledgement and Methods (page 39, line 12; page 40, line 18-21; page 48, line 22; page 53, line 1; page 59, line 6-7).

Comments on the authors' response to Reviewer 1

1. One of the major concerns from Reviewer 1 is that Foxo1 and SREBP1 pathways are regulated in opposite manners in liver. It is difficult to conceptualize the idea that CtBP2 functions to inhibit both Foxo1 and SBREBP1 pathways. The authors responds that the activity of CtBP2 is relatively maintained in both fasting and random-fed conditions. Although CtBP2 inhibits both Foxo1 and SREBP1 pathways, the mechanism is kind of

different. CtBP2 binds to Foxo1 directly and inhibits its activity. CtBP2 binds to SREBP1 through LRH1 indirectly and inhibits its activity. In my opinion, it may be true that CtBP2 does not play a key role in regulation of Foxo1 and SREBP1 pathways under physiological state, whereas other mechanisms are involved. The reason might be the key metabolites regulating CtBP2 activity, such as NADPH and fatty acyl-CoA, are maintained at a reasonable range. However, under a pathological condition (obesity), owing to the significant increase of fatty acyl-CoA, the activity of CtBP2 is inhibited. Foxo1 and SREBP1 pathways are activated, which impairs glucose and lipid homeostasis. To some extent, the role CtBP2 may explain the selective insulin resistance in diabetes.

Response: We sincerely appreciate the Reviewer #4's comprehensive understanding of our manuscript.

2. Another important question the Reviewer 1 raised is that the Foxo1 occupancy in the promoters of G6pc and Pck1 is similar in lean and HFD mice. I am not sure it is true or not, since Foxo1 protein level should be upregulated in HFD mouse livers and accordingly the Foxo1 occupancy in its target genes' promoters should be also increased. However, these results do not contradict to the rationale the authors proposed. They propose that CtBP1 directly inhibits Foxo1 activity without affecting its occupancy in the promoters of G6pc and Pck1. If so, another interesting question is how CtBP1, as a transcription repressor, inactivates Foxo1.

Response: We appreciate this comment as well. CtBP2 has been reported to remodel chromatin architecture through recruiting histone modifying enzymes, which results in transcriptional repression in most of cases. We showed increased repressive histone marks at G6pc gene promoter, where CtBP2/FoxO1 complex is located, by CtBP2 overexpression (Extended Data Fig. 7f). Thanks to this suggestion, we were able to describe a sequential event where alteration of the interaction between CtBP2 and transcription factors by metabolites leads to chromatin remodeling and transcriptional repression (page 28 line 1-page 29 line 2).

In general, the authors answered the reviewer 1's questions. The most questions of review 1 is generated from the molecular mechanisms of CtBP1. In this manuscript, the authors did

not clearly describe the molecule mechanisms of CtBP1. Thus, I suggest the authors add this part into the discuss

Response: We took this comment as a request for mechanisms of CtBP2 not CtBP1 since the Reviewer #1 asked about CtBP2 in the previous round of review. This comment kindly reminded us that metabolite-dependent interactions of CtBP2 weighed heavily in our manuscript where we have some gaps in the comprehensive molecular mechanisms linking metabolic alterations to transcriptional repression. As we stated above, we moved our supplementary discussion to the main text and added a few sentences to clarify that CtBP2 binds to transcription factors such as FoxO1 and SREBP1 where CtBP2 remodels chromatin structures, that was also experimentally supported by the Extended Data Figure 7f (page 28 line 1-page 29 line 2). Thanks to this suggestion, we were able to describe a sequential event where alteration of the interaction between CtBP2 and transcription factors by metabolites leads to chromatin remodeling and transcriptional repression.

We also added sentences to clarify data availability and our statistical analyses in the Methods section. We changed the order of the authors according to their contributions.

Thank you again for your careful consideration of our work. We look forward to hearing from you.

Sincerely,

Motohiro Sekiya, M.D., Ph.D.

Associate professor

Faculty of Medicine (Endocrinology and Metabolism)

University of Tsukuba

Reviewers' Comments:

Reviewer #3:

Remarks to the Author:

The authors have adequately addressed my concerns.

Reviewer #4:

Remarks to the Author:

My concerns were well addressed.